# LiveProteinBench: A Contamination-Free Benchmark for Assessing Models' Specialized Capabilities in Protein Science

## Abstract

In contrast to their remarkable performance on general knowledge QA, the true abilities of Large Language Models (LLMs) in tasks demanding deep, specialized reasoning, such as in protein biology, have yet to be thoroughly investigated. Current benchmarks suffer from critical deficiencies, such as data contamination due to outdated test sets, insufficient focus on essential protein-specific tasks, and a neglect of multimodal assessments. To resolve these issues, we introduce Live-ProteinBench, a contamination-free, multimodal benchmark of 12 tasks for evaluating LLM performance on protein property and function prediction. Its central innovation lies in a test set composed exclusively of proteins validated after the start of 2025, guaranteeing that the data is novel to all tested models. We benchmarked a suite of prominent general-purpose LLMs and specialized biological LLMs using both unimodal and multimodal input schemes. Our results show that: 1) General-purpose proprietary large models demonstrate superior zero-shot performance when encountering new protein data, outperforming their open-source and domain-specific counterparts by over 20% accuracy. 2) The effective use of multi-view structural information remains a significant challenge, as the inclusion of structural images often fails to provide a consistent benefit and can even degrade performance. This highlights the limitations of current models in effectively fusing information across different modalities. 3) Models' performance scales more directly with the computational cost during inference than with its parameter count, underscoring the critical role of Chain-of-Thought reasoning capabilities for protein-specific tasks. LiveProteinBench delineates the current performance frontiers for LLMs in bioinformatics and presents new challenges for the development of future multimodal foundation models for biology.

## 1 Introduction

Large Language Models (LLMs) have recently become a new paradigm for advancing protein research Madani et al. (2023); Xiao et al. (2025b). However, genuinely unlocking the secrets of life requires these models to move beyond merely processing sequence information and to demonstrate multiple advanced capabilities. These include the ability to precisely follow complex instructions from researchers He et al. (2024b), the capacity to reason by deeply integrating vast biological knowledge Xu et al. (2025), and the multimodal understanding Bhattacharya et al. (2024) to integrate information from both sequence and structure. In this context, numerous models have been developed in both academic and industrial settings, ranging from specialized models Guo et al. (2023); Lv et al. (2025) tailored for particular tasks to powerful general-purpose models Yang et al. (2025); Comanici et al. (2025) with cross-domain applications. With this proliferation of models, establishing a comprehensive and reliable benchmark to fairly and systematically evaluate their true capabilities on biological tasks has become an urgent priority.

Although some preliminary evaluation efforts have been made in academia, existing methods for fairly and reliably assessing the true capabilities of AI models generally face three core challenges. Firstly, there is a disconnect between general-purpose benchmarks and the requirements of specialized domains. Current mainstream benchmarks for LLMs Wang et al. (2024); Li et al. (2024)

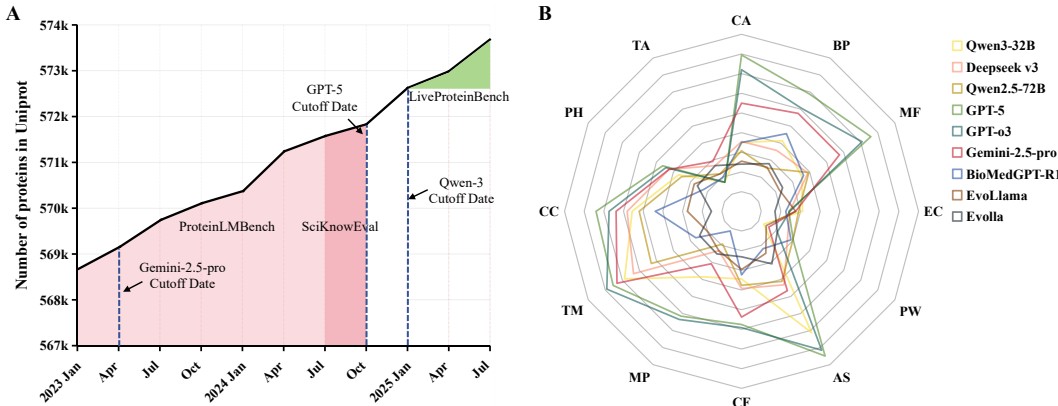

Figure 1: Overview of the LiveProteinBench Design and Model Evaluation Performance. (A) Illustration of the "Live Data" design principle. The red-shaded area represents the data range used by previous benchmarks, and the green-shaded area represents the data range used by LiveProteinBench. (B) Radar chart of the performance of leading large language models across the 12 core tasks. Each axis corresponds to a specific task, with performance scaling proportionally to the distance from the center.

primarily assess general knowledge and universal capabilities, but they severely lack an in-depth examination of core biological problems like protein sequence analysis and structure-function relationships. Therefore, they cannot be used to conduct a detailed evaluation of a model's true performance in the highly specialized field of protein science. Secondly, evaluations specialized for proteins are often susceptible to data leakage. The opaque nature of a large model's pre-training data, which usually contains immense volumes of text from the web and scientific papers, creates a high probability that test set protein sequences were part of the model's training data Feng et al. (2024); He et al. (2024a). This risk is intensified by the fact that many current protein benchmarks are derived from outdated sources or lack transparency in their creation. This makes it difficult for an evaluation to discern between a model's ability to generalize through reasoning and its capacity for simple memorization, thereby fundamentally compromising the effectiveness of the assessment Yu et al. (2025). Thirdly, existing benchmarks pay insufficient attention to inherent structural information. The three-dimensional structure of a protein has a profound impact on its function. However, existing evaluation efforts Shen et al. (2024); Jiang et al. (2025) generally neglect to assess a model's ability to integrate and understand a protein's specific structural information. Whether large general-purpose models can effectively integrate abstract sequence data with concrete structural information, as specialized models are designed to do, remains a key open question that urgently requires rigorous validation.

To address this crucial gap, we present LiveProteinBench, a new framework for the precise evaluation of protein comprehension in LLMs. It is defined by three core features that directly counteract the challenges outlined above: expert-level task design, reliable and contamination-free data sourcing, and the examination of essential multimodal capabilities. Our work incorporates the following designs to meet these objectives: First, at the level of task definition, LiveProteinBench comprises 12 professional, progressively structured tasks that cover everything from basic sequence attribute prediction to high-level functions and localizations. The philosophy behind these tasks is to probe a model's ability to reason, analyze, and make predictions from protein data, enabling a systematic assessment of the model's professional skill and breadth when tackling practical biological questions. Second, at the data source level, we ensure the benchmark's integrity through a "Live Data" construction methodology. All evaluation data consists of records generated after January 1, 2025, from authoritative and continuously updated databases such as UniProt. This time point postdates the knowledge cut-off for most leading large models. This strict temporal partitioning strategy eradicates the possibility of data contamination from the outset, ensuring a fair evaluation that genuinely measures a model's capacity for generalization and reasoning. Finally, regarding the testing protocol, in addition to providing standardized prompts and evaluation scripts to guarantee reproducibility, we have introduced 3D protein structure as a new input modality. As general-purpose

multimodal models typically only accept images, we convert structures into six-view projection images. This allows us to assess a key capability of these large models—their ability to fuse multimodal biological information—by comparing their performance with and without the structural input.

To summarise, the main contributions of this work are as follows:

- We developed LiveProteinBench, the first contamination-free, multi-task, and multimodal benchmark for protein science. Its 12 core tasks and strict temporal data split provide a reliable benchmark for holistically assessing the real-world capabilities of large models.
- We conducted extensive evaluation on over 10 prominent Large Language Models, spanning both general-purpose and domain-specific architectures. Our findings reveal that leading general-purpose models exhibit significant zero-shot reasoning potential on complex biological tasks.
- We systematically investigated the multimodal capabilities of models for protein understanding. Our results uncover a critical challenge: for current models, supplementing sequence data with structural information may degrade performance, indicating a major bottleneck in effective multimodal information fusion.
- Our analysis offers a new perspective on scaling laws in bioinformatics. We demonstrate that for specialized protein tasks, a model's success is more strongly correlated with its "Chain-of-Thought" reasoning ability than with its parameter count, suggesting that future advancements may depend more on algorithmic and reasoning improvements.

## 2 RELATED WORK

### 2.1 GENERAL-PURPOSE AND PROTEIN-SPECIFIC LARGE LANGUAGE MODELS

Recent years have seen rapid advancements in the development of Large Language Models (LLMs). One line of research is led by models such as the GPT-4 Achiam et al. (2023), Gemini Team et al. (2023), Llama Dubey et al. (2024), and Qwen Bai et al. (2023) series, which are at the forefront of technology due to their powerful Generality. Pre-trained on massive, diverse datasets, these models exhibit outstanding performance in instruction following, logical reasoning, and multimodal comprehension. As general-purpose problem-solving engines, their application in specialized fields like protein science is focused on transferring this general intelligence to specific scientific discovery tasks. Consequently, a key challenge is to determine whether their generalist training is sufficient to confer deep, specialized proficiency in areas like protein biology. To address the domain knowledge gap of general-purpose models, a second research direction is the development of specialized models for protein science, defined by their core trait of Specialization. These models generally employ two strategies to deepen their understanding of the language of life. The first strategy is continued pre-training or instruction fine-tuning on domain-specific data like protein sequences, structures, and scientific texts, as exemplified by models such as EvoLlama Liu et al. (2024b), Evolla Zhou et al. (2025), STELLA Xiao et al. (2025a) and Prot2Text-V2 Fei et al. (2025). The second involves creating in-domain dialogue or QA models; for instance, ProteinChat Guo et al. (2023) and ProteinGPT Xiao et al. (2024) are fine-tuned on protein-centric question-answer pairs. Additionally, other efforts like BioMedGPT Luo et al. (2023) aim to create models capable of processing diverse multimodal biomedical data, including proteins, genes, and literature, in a unified manner. Despite their different design approaches, both types of models are highly dependent on existing public databases and scientific literature for training. This creates a shared, fundamental challenge: the risk of data contamination is extremely high when evaluations are performed using traditional benchmarks derived from these same public knowledge sources. This highlights two urgent needs: first, a contamination-free evaluation framework is essential for an unbiased assessment of any model's true capabilities. Second, a professional, multi-task benchmark is required to systematically probe and compare the distinct strengths and potential limitations of both general-purpose and specialized models.

### 2.2 PROTEIN BENCHMARKS

Early protein evaluation benchmarks focused primarily on assessing Protein Foundation Models (PFMs). For instance, TAPE Rao et al. (2019) set the initial standard for protein transfer learning. Subsequently, PEER Xu et al. (2022) significantly expanded the breadth of evaluation for protein

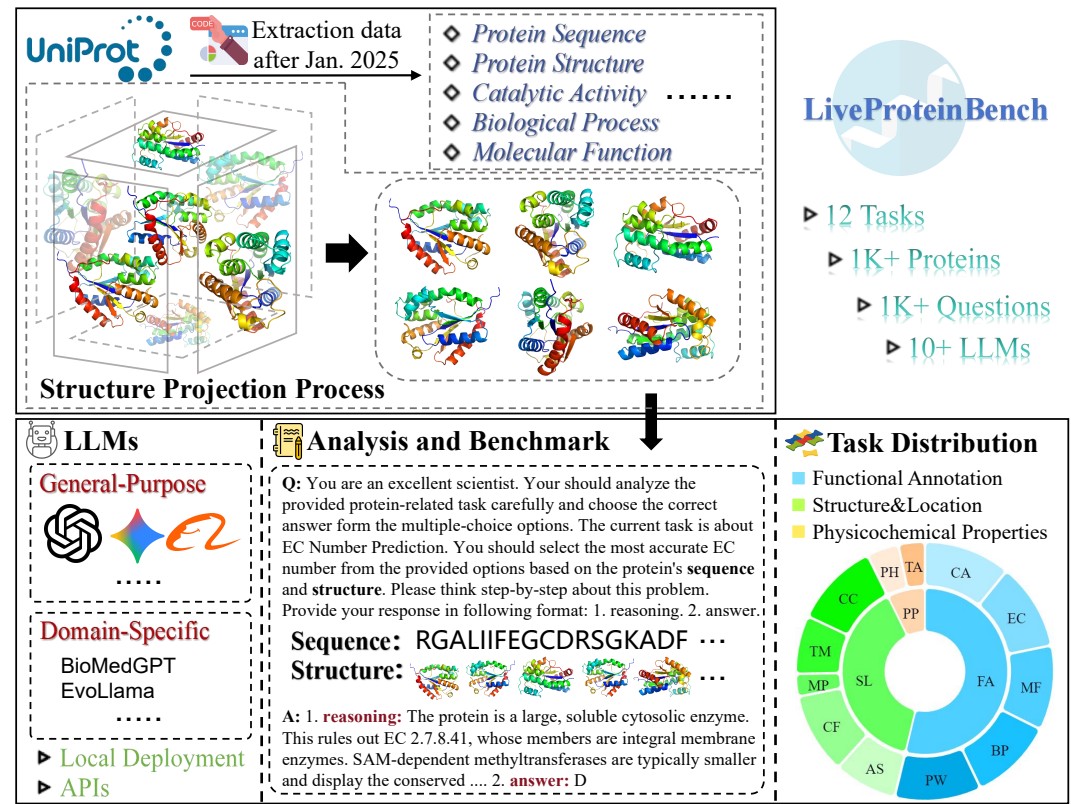

Figure 2: The framework of LiveProteinBench, which consists of a live data collection pipeline from UniProt, a structure projection process for multimodal inputs, and 12 diverse tasks covering three key categories. The evaluation data in the benchmark is strictly filtered for proteins released after January 2025 to ensure a contamination-free assessment.

sequence understanding by providing a multi-task benchmark with 14 tasks. More recently, comprehensive benchmarks like PFMBench Gao et al. (2025) and ProteinBench Ye et al. (2024) further enlarged the task sets to cover a wider range of protein functions and properties, while CARE Yang et al. (2024) concentrated on the specific yet critical tasks of enzyme classification and retrieval. As the capabilities of LLMs have grown, the focus of evaluation has begun to shift from traditional PFMs to general-purpose LLMs. For example, Biology-Instructions He et al. (2024a) aims to assess a model's ability to follow complex biological experimental instructions, SciCUEval Yu et al. (2025) provides an in-depth evaluation of the scientific capabilities of LLMs from the perspective of contextual understanding of scientific literature, and BioLLMBench Mangul et al. (2024) presents a comprehensive assessment of three widely-used LLMs (GPT-4, Bard, and LLaMA) across 36 distinct tasks within the domain of bioinformatics. Nevertheless, these LLM-focused benchmarks suffer from two key limitations. First, while structural information is central to protein biology, existing LLM benchmarks have largely overlooked the evaluation of multimodal capabilities, focusing almost exclusively on how models interpret sequence-based text. This leaves a critical gap in understanding how these models integrate diverse biological data types. Second, and more critically, their test sets are typically drawn from historical data, creating a high risk of data contamination, as this information may have been part of the pre-training corpora of the models being evaluated. This makes it difficult to distinguish between genuine reasoning and mere memorization.

## 3 LIVEPROTEINBENCH

In this section, we introduce the composition of LiveProteinBench, covering the dataset construction methodology and our assessment process.

## 3.1 Evalaution Protocol

Table 1: Task composition and distribution of the LiveProteinBench.

| Class | Task | Number | Abbreviation |
|---|---|---|---|
| **Functional Annotation (FA)** | Catalytic Activity | 200 | CA |
| | EC Number | 200 | EC |
| | Molecular Function | 195 | MF |
| | Biological Process | 193 | BP |
| | Pathway | 200 | PW |
| **Structure&Location (SL)** | Active Site | 146 | AS |
| | Cofactor | 186 | CF |
| | Motif Position | 52 | MP |
| | Transmembrane | 134 | TM |
| | Cellular Component | 196 | CC |
| **Physicochemical Properties (PP)** | Optimal pH | 54 | PH |
| | Thermal Adaptation | 41 | TA |

### 3.1.1 Task Composition.

We have carefully designed 12 tasks to comprehensively evaluate a model's understanding of different protein attributes. To ensure absolute objectivity and reproducibility, the ground truth for all tasks is programmatically extracted directly from experimentally validated annotations in the source databases. This design eliminates any ambiguity or noise that could be introduced by manual labeling or text summarization, guaranteeing that our evaluation is precise and verifiable. The full composition and distribution of these tasks are detailed in Table 1; detailed definitions are provided in Appendix G.

**Functional Annotation (FA).** This class of tasks is designed to evaluate a model's ability to comprehend a protein's biological role and function. It covers a spectrum of annotations, from the specific chemical reactions a protein catalyzes—assessed by Catalytic Activity (CA) and its formal EC Number—to its broader involvement in cellular life. The latter is evaluated using three tasks based on established biological frameworks: Molecular Function (MF) and Biological Process (BP) from the Gene Ontology (GO) consortium, and the metabolic or signaling Pathway (PW) in which the protein participates.

**Structure & Location (SL).** This category assesses the model's understanding of a protein's physical attributes and its subcellular localization. The tasks probe both local and global features. On a local level, we evaluate the ability to identify key functional regions, including the Active Site (AS), Cofactor (CF), and conserved Motif Positions (MP). On a global and contextual level, we test for the identification of Transmembrane (TM) regions and the prediction of the protein's correct Cellular Component (CC), which is the third main branch of the Gene Ontology.

**Physicochemical Properties (PP).** This class focuses on the model's capacity to predict global properties that emerge from the protein's entire amino acid sequence, rather than from a single site. These properties dictate the environmental conditions under which a protein can function. The tasks include predicting the protein's Optimal pH (PH) for activity and its Thermal Adaptation (TA) class, which requires a holistic understanding of the protein's stability.

### 3.1.2 Task Formulation and Evaluation Metric

To achieve an objective, reproducible, and highly discriminative evaluation, we have uniformly designed all tasks in a multiple-choice question (MCQ) format. In each test instance, the model is required to select a single correct answer from a set of candidate options based on the input protein information. To ensure that all models are evaluated under fair and consistent conditions, we designed a standardized prompt structure. This structure is composed of two parts, a System Prompt and a User Prompt, and the final input received by the model is a combination of these two. Given that all tasks are designed in a single-choice question format, we adopt accuracy, which is defined as the proportion of correctly answered questions, as the core evaluation metric.

## 3.2 DATASET CONSTRUCTION

The construction of the LiveProteinBench dataset is guided by three core principles to ensure its evaluation is rigorous, fair, and forward-looking: professionalism, achieved through expertly designed tasks and programmatically verifiable ground truths; data non-contamination, guaranteed by a strict "Live Data" filtering methodology; and multimodality, incorporated by supplementing each protein sequence with 3D structural views.

**Data Sourcing and Filtering.** To ensure that all evaluated models have not previously seen the test data, we established a strict temporal filtering criterion. We selected experimentally validated proteins from UniProtuni (2025) that were first publicly released after January 1, 2025. This standard was strictly enforced by cross-validating the creation dates of database entries with their associated publication dates. This method fundamentally prevents data leakage and ensures the fairness of the evaluation. Furthermore, to ensure the long-term relevance of the benchmark, we established an automated update pipeline. This system periodically scans source databases like UniProt to automatically identify and incorporate the latest protein entries that meet our strict criteria. This "live" mechanism ensures our benchmark continuously stays ahead of the training data cut-offs of future LLMs and dynamically reflects the latest advances in protein science.

**Data Contribution.** For multimodal evaluation, we implemented a **Structure Projection Process** to generate structural images for each protein in our collected dataset. When a protein structure was available, we retrieved it from the AlphaFold DB Varadi et al. (2024); otherwise, we used AlphaFold2 Jumper et al. (2021) to generate the top-ranked structure. We then employed the PyMOL DeLano et al. (2002) rendering tool to create 2D structural snapshots from six standard orthogonal views (front, back, left, right, top, and bottom) to provide comprehensive three-dimensional structural information.

## 4 EXPERIMENTS

### 4.1 EXPERIMENTAL MODELS

To comprehensively evaluate the application potential of current large language models in protein science, our study includes over 10 mainstream and representative models. The selection is designed to span the spectrum from general-purpose knowledge to domain-specific expertise and to systematically investigate the relationship between general capabilities and specialized knowledge. The models are divided into two main categories:

**General-Purpose LLMs.** This category aims to establish a strong performance baseline and includes leading closed-source and open-source models. Closed-Source Models: We selected OpenAI's GPT series, Google's Gemini 2.5 Pro Comanici et al. (2025) and Gemini 2.0 Flash Team et al. (2023), and Anthropic's Claude 3.7 Sonnet. Open-Source Models: We selected Deepseek-v3 Liu et al. (2024a), Deepseek-r1 Guo et al. (2025), InternVL3 Zhu et al. (2025), Meta's Llama-3.1 Dubey et al. (2024) and Alibaba's Qwen series (e.g., Qwen2.5-72B) Yang et al. (2025); Bai et al. (2025).

**Domain-Specific Models.** This category aims to evaluate models that are specialized for the protein domain, including Evolla Zhou et al. (2025), EvoLlama Liu et al. (2024b) and BioMedGPT Luo et al. (2023). Unlike general-purpose models, these models are typically pre-trained or fine-tuned on corpora containing a vast amount of literature from biology, chemistry, and medicine, as well as patents and specialized databases.

### 4.2 MAIN RESULTS

The overall and per-task scores on LiveProteinBench are presented in Table 2. Our results systematically reveal a performance hierarchy among current LLMs when processing novel protein data, leading to several key findings:

**General-Purpose Models Outperform Specialized Models.** The results show that the overall performance of large-scale general-purpose models (LLMs and MLLMs) is significantly superior to that of small-scale specialized models (SLLMs). For example, in terms of average performance across all tasks, no SLLM scored above 32%, which is lower than all general-purpose models except for Qwen2.5-VL-32B and InternVL3-78B. This performance gap is not solely attributable to

Table 2: Overall performance of all models on LiveProteinBench. Performance is measured by accuracy (%). ♣ and ♡ denote sequence-only and sequence+structure input modalities, respectively. The **best** and suboptimal results are labeled with bold and underlined.

| Model | CA | BP | MF | EC | PW | AS | CF | MP | TM | CC | PH | TA | AVG |
|---|---|---|---|---|---|---|---|---|---|---|---|---|---|
| **LLMs** | | | | | | | | | | | | | |
| ♣Qwen3-32B | 35.00 | 41.45 | 35.38 | **31.00** | 13.00 | 71.23 | 34.41 | 38.46 | 68.66 | 55.61 | 37.04 | 17.07 | 40.23 |
| ♣Deepseek-v3 | 35.50 | 35.75 | 38.97 | 24.50 | 27.00 | 43.15 | 39.25 | 23.08 | 63.43 | 58.16 | 42.59 | 26.83 | 38.95 |
| ♣Deepseek-r1 | 45.50 | 37.31 | 46.67 | 22.50 | 8.50 | 64.38 | 32.80 | 38.46 | 49.25 | 55.61 | 42.59 | 12.20 | 38.62 |
| ♣Llama3.3-70B | 36.00 | 32.12 | 40.51 | 21.50 | 27.50 | 36.99 | 41.40 | 23.08 | 54.48 | 59.69 | 46.30 | 19.51 | 37.67 |
| ♣Qwen2.5-72B | 30.77 | 25.39 | 39.49 | 29.00 | 30.00 | 41.10 | 37.63 | 19.23 | 52.99 | 52.04 | 35.19 | 21.95 | 35.98 |
| ♣Qwen2.5-32B | 31.00 | 29.02 | 35.38 | 24.00 | **30.50** | 37.67 | 29.03 | 19.23 | 61.94 | 52.04 | **50.00** | 17.07 | 35.28 |
| **MLLMs** | | | | | | | | | | | | | |
| ♡GPT-5 | **79.50** | **68.91** | **75.90** | 24.00 | **30.50** | **84.93** | 57.53 | 61.54 | 75.37 | **73.98** | 46.30 | 17.07 | **60.65** |
| ♡o3 | 72.00 | 60.62 | 70.77 | 24.50 | 21.00 | 81.51 | **59.14** | **63.46** | **79.10** | 67.35 | 44.44 | 17.07 | 56.48 |
| ♡Gemini-2.5-pro | 55.00 | 57.51 | 57.44 | 27.50 | 16.00 | 46.58 | 53.76 | 30.77 | 73.13 | 63.78 | 42.59 | 29.27 | 47.97 |
| ♡Claude-3.7-sonnet | 66.50 | 42.49 | 52.82 | 23.00 | 29.00 | 56.85 | 58.06 | 23.08 | 72.39 | 53.57 | 35.19 | 21.95 | 47.58 |
| ♡Gemini-2.0-flash | 40.00 | 32.12 | 44.62 | 24.00 | 29.50 | 33.56 | 39.78 | 28.85 | 72.39 | 58.16 | 29.63 | **36.59** | 39.84 |
| ♡GPT-4o | 38.50 | 38.86 | 41.54 | 27.00 | 24.50 | 36.99 | 30.11 | 28.85 | 53.73 | 46.94 | 44.44 | 14.63 | 36.45 |
| ♡Qwen2.5-VL-32B | 29.50 | 24.35 | 24.62 | 24.00 | 27.64 | 34.48 | 39.33 | 21.15 | 52.24 | 33.67 | 42.59 | 14.63 | 30.98 |
| ♡InternVL3-78B | 30.00 | 25.91 | 26.15 | 24.50 | 19.00 | 24.66 | 33.87 | 21.15 | 32.09 | 22.96 | 27.78 | 19.51 | 26.10 |
| **SLLMs** | | | | | | | | | | | | | |
| ♣BioMedGPT-R1 | 35.00 | 45.60 | 36.41 | 22.50 | 29.00 | 21.92 | 32.26 | 11.54 | 26.87 | 43.88 | 22.22 | 19.51 | 31.83 |
| ♡EvoLlama | 25.50 | 25.91 | 22.05 | 27.50 | 14.50 | 24.66 | 29.57 | 23.08 | 20.15 | 27.55 | 27.78 | 21.95 | 24.26 |
| ♡Evolla | 24.00 | 27.98 | 24.10 | 17.00 | 21.00 | 30.82 | 23.12 | 25.00 | 24.63 | 15.31 | 25.93 | 26.83 | 22.98 |

differences in model scale. Even when comparing models of a similar parameter size (e.g., in the 7B-13B range), general-purpose models often maintain a performance advantage, suggesting that the robust instruction-following and reasoning capabilities acquired during their pre-training are effectively transferable to specialized domains. We observed during testing that specialized models were more prone to failing to generate valid or correctly formatted responses, which likely contributes to this performance gap. For a detailed error analysis and statistics on valid response rates, please refer to Appendix F.

**Dominance of Closed-Source Models.** Within the general-purpose model category, the most significant performance difference lies between closed-source and open-source models. The evaluation results clearly show that top-tier closed-source models, particularly GPT-5, are substantially ahead of all open-source models in performance. Even the lowest-performing closed-source model, Gemini 2.0 Flash, achieved an average accuracy across all tasks that is comparable to that of the best-performing open-source model, Qwen-3. This significant performance advantage is primarily due to the larger model scales, higher-quality and more diverse training data, and more advanced alignment techniques of these closed-source models.

**Performance Disparity Across Tasks.** A cross-task comparison reveals inherent differences in task complexity and highlights commonalities in what current models can and cannot do well. Models generally excel at tasks involving local feature identification and static property classification, such as AS, CC, and CA. The common characteristic of these tasks is that the answer often corresponds to a specific, localized region of the protein or a discrete categorical label, relying more on pattern recognition. In contrast, all models face significant challenges on tasks that require holistic reasoning or an understanding of dynamic mechanisms, such as TA and EC. TA requires comprehending subtle, distributed patterns across the entire sequence that determine overall stability, while EC prediction demands an understanding of the dynamic process of enzyme catalysis. On the EC task, even the best-performing model (Qwen3-32B) achieved an accuracy of only 31.00%. This suggests that reasoning about dynamic processes and complex mechanisms is a key bottleneck for current models and a critical direction for future development

## 4.3 ANALYSIS

**Task Correlations.** The correlation heatmap in Figure 3 (A) reveals a complex and insightful network of relationships in how models perform across different biological reasoning tasks. On one hand, we observe strong coupling between functionally and structurally related tasks; for instance, the three branches of the Gene Ontology (MF, BP, and CC) are highly inter-correlated. A particularly interesting finding is the tight link between MP and AS, suggesting that models learn an

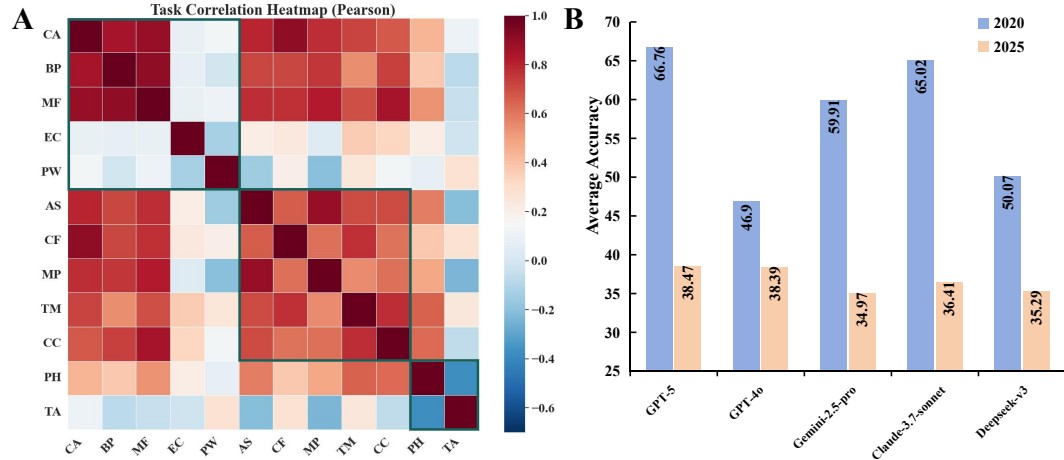

Figure 3: Benchmark Analysis. (A) Pearson correlation heatmap of the 12 tasks in LiveProtein-Bench. The green boxes highlight three major categories of related tasks. (B) Performance comparison of top LLMs on data from 2020 versus our contamination-free LiveProteinBench.

effective heuristic of using conserved motifs to identify functional sites. On the other hand, some seemingly related tasks show weaker links, such as EC and PW. This reflects their different levels of abstraction. This distinction is most pronounced for the physicochemical tasks (PH and TA), which are almost entirely independent of all others, clearly indicating that reasoning about global protein properties is a unique capability, separate from functional annotation.

**Temporal Validation of Data Contamination.** We performed a temporal analysis to rigorously validate our "Live Data" approach. For this validation, we curated a set of protein entries from UniProt that were first made public in 2020, a time frame that is certain to be included in the training data of all models. From this collection, we randomly sampled a number of questions equivalent to that of our main experiment. We then evaluated a selection of models on the five tasks where they exhibited the poorest performance, including EC, PW, TM, PH and TA. As illustrated in the Figure 3 (B), the results show a marked decline in performance for these models as the publication date of the data advances toward the present. This demonstrates that LiveProteinBench measures a model's true generalization and reasoning abilities, rather than its memorization of old knowledge that might be present in the training data.

**Challenges in Multimodal Protein Understanding.** Our research uncovers key bottlenecks in how current multimodal models comprehend protein structural information. First, contrary to the intuition that more information should yield better results, introducing a single protein structure image fails to provide a consistent or significant performance benefit. As shown in Figure 4 (A), the effect is inconsistent across different models: for some, like Claude-3.7-sonnet, performance even dropped noticeably from 51.47% to 45.77%, while for others, the difference was negligible. This suggests that the visual encoders of current models, primarily trained on natural images, struggle to interpret highly specialized scientific images, failing to extract beneficial information and potentially introducing noise that interferes with the core sequence data. Second, fusing multi-view information presents another major hurdle. Figure 4 (B), providing all six structural views leads to lower performance than using any single view. This indicates that while sufficient information is present across the different views, the model lacks the ability to effectively fuse these 2D images into a coherent 3D concept. Enabling models to understand and fuse multi-view scientific data is a critical direction for future multimodal development.

**Reasoning is the Key Performance Driver.** Our analysis suggests that the scaling law in this domain manifests more through the depth of reasoning than through raw parameter count. On one hand, the Chain-of-Thought (CoT) process is critical. As shown in Figure 4 (C), activating CoT mode led to significant and consistent performance gains across all models that support this feature, with the Qwen3-32B model's accuracy increasing by over five percentage points. This demonstrates that an explicit "thinking" step is effective in unleashing the full potential of these models. On

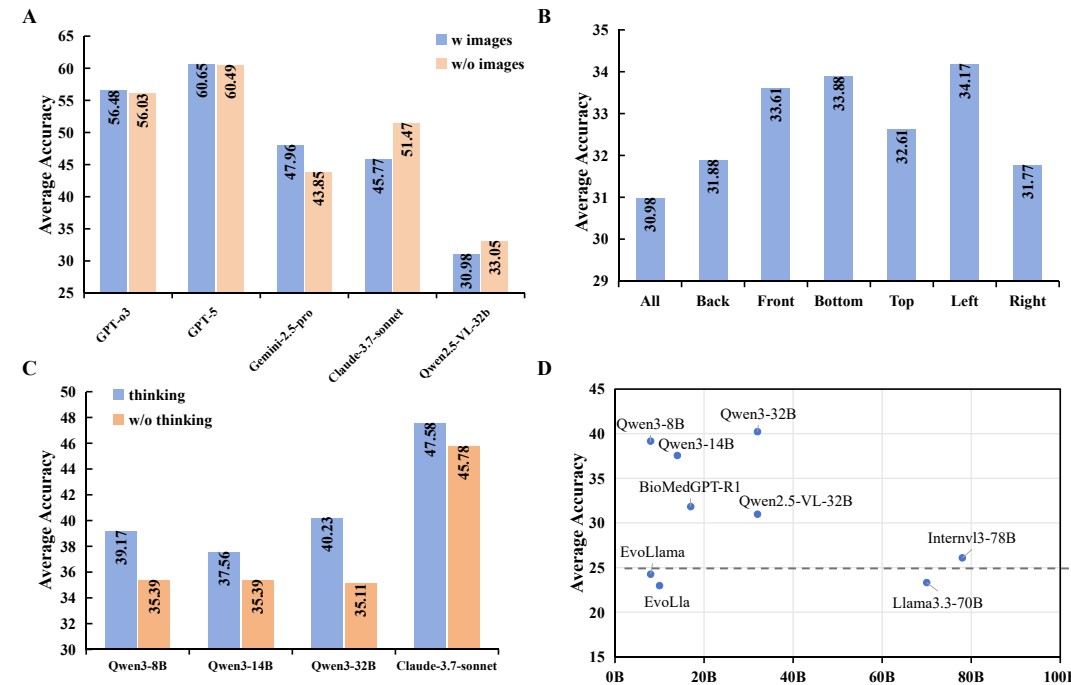

Figure 4: A Comprehensive Analysis of LLM Performance. (A) The Anomaly of Multimodal Input. (B) The Fusion Bottleneck of Multi-view Structural Images. (C) The Key Role of Chain-of-Thought Reasoning. (D) The Non-linear Relationship Between Model Scale and Performance.

the other hand, simply increasing parameter size yields diminishing returns. As seen in Figure 4 (D), within the Qwen3 series, the parameter count quadrupled from 8B to 32B, yet accuracy almost stagnated. This indicates that for complex protein tasks, factors like architectural design, training data quality, and fine-tuning strategies have surpassed parameter expansion in importance.

# 5 CONCLUSION

We present LiveProteinBench, a contamination-free, multi-task, and multimodal benchmark designed to assess the real-world capabilities of Large Language Models (LLMs) in protein science. This benchmark aims to solve three core challenges in existing evaluation methods: their disconnect from the specialized domain, the prevalent risk of data contamination, and the neglect of multimodal capabilities. The primary innovation of LiveProteinBench is its "Live Data" construction method, where all evaluation data consists of protein records released after January 1, 2025. This date is later than the knowledge cut-off of all mainstream large models, fundamentally eliminating the possibility of data contamination. The framework uses 12 carefully designed, professional tasks to systematically probe a model's skills in solving real-world biological problems. Unlike previous benchmarks that only focus on sequence information, LiveProteinBench introduces projected images of 3D protein structures as a new input modality to assess the model's ability to fuse multimodal biological information. Through the extensive evaluation of over 10 mainstream general-purpose and specialized models, this paper presents a series of key findings that provide significant guidance for the field.

**Limitations and Future Work.** The 12 tasks in LiveProteinBench primarily focus on the intrinsic properties and functions of single proteins. Our benchmark does not currently cover system-level tasks, such as protein-protein interactions, protein-small molecule docking, or predicting the functional effects of mutations. We plan to expand the scope of LiveProteinBench in the future by introducing more evaluation tasks related to systems biology and protein engineering. This will place higher demands on the models' reasoning abilities for more macroscopic and dynamic biological problems.

ETHICS STATEMENT

This work adheres to the ICLR Code of Ethics. In this study, no human subjects or animal experimentation was involved. All datasets used, including Uniprot, were sourced in compliance with relevant usage guidelines, ensuring no violation of privacy. We have taken care to avoid any biases or discriminatory outcomes in our research process. No personally identifiable information was used, and no experiments were conducted that could raise privacy or security concerns. We are committed to maintaining transparency and integrity throughout the research process.

REPRODUCIBILITY STATEMENT

We have made every effort to ensure that the results presented in this paper are reproducible. All code has been submitted in the Supplementary Material to facilitate replication and verification. The experimental setup, including training steps, model configurations, and hardware details, is described in detail in the paper. We believe these measures will enable other researchers to reproduce our work and further advance the field.

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

## A  THE USE OF LARGE LANGUAGE MODELS

During the preparation of this manuscript, we utilized Gemini-2.5-pro as an assistive tool for language enhancement. The primary use of these models was to improve the grammar, clarity, and readability of the text. All scientific ideas, experimental results, and conclusions were conceived and written by the human authors, who retain full responsibility for the final content of this paper.

## B  HARDWARE AND SOFTWARE ENVIRONMENT

To ensure the reproducibility of our results for the locally deployed open-source models, all evaluations were conducted within a unified computational environment. The evaluation server was equipped with 8x NVIDIA H100 (80GB VRAM) GPUs, powered by AMD EPYC 9654 96-Core Processor and 3TB of DDR4 RAM. The software stack consisted of Ubuntu 22.04 LTS, running Python 3.10.12 with CUDA 12.2, and utilized key libraries including PyTorch (v2.4.1), Transformers (v4.56.1), and Accelerate (v1.10.0).

## C  EXPERIMENTAL MODELS

To comprehensively evaluate the application potential of current large language models in protein science, our study includes over 10 mainstream and representative models. As detailed in Table 3, these models are divided into three main categories to systematically investigate the relationship between general capabilities and specialized knowledge.

**General-Purpose LLMs.** This category aims to establish a strong performance baseline and includes leading closed-source and open-source models.

- Closed-Source Models: We selected OpenAI's GPT series (GPT-o3, GPT-5, GPT-4o), Google's Gemini series (Gemini-2.0-flash, Gemini-2.5-pro), and Anthropic's Claude-3.7-sonnet.
- Open-Source Models: We selected Deepseek-V3 (671B), Deepseek-R1 (671B), Qwen2.5-72B (72B), Qwen2.5-32B (32B), Qwen3-32B (32B), Llama3.3-70B (70B), Qwen2.5-VL-32B (32B) and InternV3-78B (78B).

**Domain-Specific Models.** The selected models are Evolla (10B), EvoLlama (8B), and BioMedGPT-R1 (17B).

## D  ADDITIONAL RESULTS OF ANALYSIS

### D.1  TEMPORAL VALIDATION OF DATA CONTAMINATION.

We compare the performance of various models under two conditions: evaluating them on a traditional dataset with data from 2020, compared to our contamination-free LiveProteinBench. As presented in Table 4, our results offer a comprehensive overview of how data contamination influences model accuracy.

### D.2  MULTIMODAL ABLATION STUDY RESULTS

We compare the performance of various models under two conditions: using both protein sequence and structural images as input, compared to using sequence-only input. As presented in Table 5, our results offer a comprehensive overview of how multimodal information influences model accuracy.

### D.3  MODEL PERFORMANCE WITHOUT A REASONING STEP

In this section, we present the results of an ablation study on models with a toggleable reasoning mechanism, such as the Qwen3 series and Claude 3.5 Sonnet. As shown in Table 6, we analyze the performance difference between activating and deactivating the "thinking" mode to evaluate the impact of the model's reasoning process.

Table 3: Models in LiveProteinBench.

| Model | Params | Expertise | Access | Code |
|-------|--------|-----------|--------|------|
| **LLMs** | | | | |
| Deepseek-V3 | 671B | General tasks | Open | API |
| Deepseek-R1 | 671B | General tasks | Open | API |
| Qwen2.5-72B | 72B | General tasks | Open | HF |
| Qwen2.5-32B | 32B | General tasks | Open | HF |
| Qwen3-32B | 32B | General tasks | Open | HF |
| Llama3.3-70B | 70B | General tasks | Open | HF |
| **MLLMs** | | | | |
| GPT-o3 | Unknown | General tasks | Close | API |
| GPT-5 | Unknown | General tasks | Close | API |
| GPT-4o | Unknown | General tasks | Close | API |
| Gemini-2.0-flash | Unknown | General tasks | Close | API |
| Gemini-2.5-pro | Unknown | General tasks | Close | API |
| Claude-3.7-sonnet | Unknown | General tasks | Close | API |
| Qwen2.5-VL-32B | 32B | General tasks | Open | HF |
| InternV3-78B | 78B | General tasks | Open | HF |
| **SLLMs** | | | | |
| Evolla | 10B | Protein tasks | Open | HF |
| EvoLlama | 8B | Protein tasks | Open | HF |
| BioMedGPT-R1 | 17B | Protein and DNA task | Open | HF |

Table 4: Performance comparison of top LLMs on data from 2020 versus our contamination-free LiveProteinBench. Performance is measured by accuracy (%).

| Model | Time | EC | PW | TM | PH | TA | AVG |
|-------|------|-----|-----|-----|-----|-----|-----|
| GPT-5 | 2020 | 65.50 | 48.00 | 95.50 | 81.48 | 51.22 | 66.76 |
| | 2025 | 24.00 | 30.50 | 75.37 | 46.30 | 17.07 | 38.47 |
| GPT-4o | 2020 | 43.50 | 38.00 | 63.43 | 53.70 | 43.90 | 46.90 |
| | 2025 | 27.00 | 24.50 | 53.73 | 44.44 | 14.63 | 32.59 |
| Gemini-2.5-pro | 2020 | 58.50 | 46.50 | 80.50 | 70.37 | 51.22 | 59.91 |
| | 2025 | 27.50 | 16.00 | 73.13 | 42.59 | 29.27 | 34.97 |
| Claude-3.7-sonnet | 2020 | 67.50 | 46.50 | 90.29 | 68.52 | 56.10 | 65.02 |
| | 2025 | 23.00 | 29.00 | 72.39 | 35.19 | 21.95 | 36.41 |
| Deepseek-v3 | 2020 | 43.50 | 40.50 | 72.39 | 61.11 | 41.46 | 50.07 |
| | 2025 | 24.50 | 27.00 | 63.43 | 42.59 | 26.83 | 35.29 |

Table 5: Performance of multimodal large language models with sequence-only input. Performance is measured by accuracy (%).

| Model | CA | BP | MF | EC | PW | AS | CF | MP | TM | CC | PH | TA | AVG |
|-------|-----|-----|-----|-----|-----|-----|-----|-----|-----|-----|-----|-----|-----|
| **w structure** | | | | | | | | | | | | | |
| GPT-5 | 79.50 | 68.91 | 75.90 | 24.00 | 30.50 | 84.93 | 57.53 | 61.54 | 75.37 | 73.98 | 46.30 | 17.07 | 60.65 |
| GPT-o3 | 72.00 | 60.62 | 70.77 | 24.50 | 21.00 | 81.51 | 59.14 | 63.46 | 79.10 | 67.35 | 44.44 | 17.07 | 56.48 |
| Gemini-2.5-pro | 55.00 | 57.51 | 57.44 | 27.50 | 16.00 | 46.58 | 53.76 | 30.77 | 73.13 | 63.78 | 42.59 | 29.27 | 47.97 |
| Claude-3.7-sonnet | 66.50 | 42.49 | 52.82 | 23.00 | 29.00 | 56.85 | 58.06 | 23.08 | 72.39 | 53.57 | 35.19 | 21.95 | 47.58 |
| Gemini-2.0-flash | 40.00 | 32.12 | 44.62 | 24.00 | 29.50 | 33.56 | 39.78 | 28.85 | 72.39 | 58.16 | 29.63 | 36.59 | 39.84 |
| GPT-4o | 38.50 | 38.86 | 41.54 | 27.00 | 24.00 | 36.99 | 30.11 | 28.85 | 53.73 | 46.94 | 44.44 | 14.63 | 36.45 |
| Qwen2.5-VL-32B | 29.50 | 24.35 | 24.62 | 24.00 | 27.64 | 34.48 | 39.33 | 21.15 | 52.24 | 33.67 | 42.59 | 14.63 | 30.98 |
| InternVL3-78B | 30.00 | 25.91 | 26.15 | 24.50 | 19.00 | 24.66 | 33.87 | 21.15 | 32.09 | 22.96 | 27.78 | 19.51 | 26.10 |
| **w/o structure** | | | | | | | | | | | | | |
| GPT-5 | 78.00 | 69.95 | 70.26 | 21.00 | 33.00 | 86.99 | 60.22 | 59.62 | 76.12 | 73.47 | 50.00 | 19.51 | 60.49 |
| GPT-o3 | 73.50 | 61.66 | 63.08 | 19.00 | 29.50 | 83.56 | 54.84 | 63.46 | 76.12 | 67.86 | 38.89 | 19.51 | 56.04 |
| Gemini-2.5-pro | 51.50 | 51.81 | 60.00 | 26.00 | 15.50 | 39.73 | 40.86 | 26.92 | 60.45 | 61.22 | 46.30 | 26.83 | 43.85 |
| Claude-3.7-sonnet | 69.00 | 53.89 | 53.33 | 26.00 | 32.00 | 53.42 | 60.75 | 26.92 | 70.90 | 66.33 | 33.33 | 36.59 | 51.47 |
| Gemini-2.0-flash | 44.00 | 38.34 | 46.67 | 25.00 | 37.00 | 36.30 | 46.24 | 32.69 | 65.67 | 61.73 | 40.74 | 24.39 | 43.07 |
| GPT-4o | 33.00 | 43.52 | 42.05 | 27.00 | 26.00 | 36.30 | 40.86 | 23.08 | 55.97 | 52.55 | 42.59 | 24.39 | 38.39 |
| Qwen2.5-VL-32B | 31.50 | 25.39 | 27.18 | 22.50 | 33.50 | 41.78 | 29.57 | 25.00 | 44.03 | 47.96 | 42.59 | 29.27 | 33.05 |
| InternVL3-78B | 28.50 | 19.69 | 22.05 | 22.00 | 18.00 | 24.66 | 25.81 | 19.23 | 25.37 | 28.06 | 22.22 | 17.07 | 23.37 |

Table 6: Model performance without the reasoning step. Performance is measured by accuracy (%).

| Model | Mode | CA | BP | MF | EC | PW | AS | CF | MP | TM | CC | PH | TA | AVG |
|---|---|---|---|---|---|---|---|---|---|---|---|---|---|---|
| Qwen3-8B | w thinking | 33.00 | 35.23 | 43.59 | 26.50 | 15.00 | 51.37 | 39.25 | 26.92 | 60.45 | 63.78 | 48.15 | 19.51 | 39.17 |
| | w/o thinking | 31.50 | 33.68 | 6.92 | 24.50 | 25.50 | 34.25 | 40.86 | 23.08 | 47.01 | 53.06 | 46.30 | 14.63 | 35.39 |
| Qwen3-14B | w thinking | 32.00 | 37.31 | 35.38 | 24.50 | 11.50 | 58.22 | 36.02 | 30.77 | 57.46 | 61.73 | 44.44 | 19.51 | 37.56 |
| | w/o thinking | 35.50 | 33.68 | 31.28 | 26.00 | 22.50 | 23.97 | 38.17 | 21.15 | 56.72 | 53.57 | 59.26 | 29.27 | 35.39 |
| Qwen3-32B | w thinking | 31.00 | 29.02 | 35.38 | 24.00 | 30.50 | 37.67 | 29.03 | 19.23 | 61.94 | 52.04 | 50.00 | 17.07 | 40.23 |
| | w/o thinking | 30.00 | 37.82 | 36.41 | 26.00 | 22.50 | 30.14 | 34.41 | 34.62 | 53.73 | 50.00 | 40.74 | 29.27 | 35.11 |
| Claude-3.7-sonnet | w thinking | 66.50 | 42.49 | 52.82 | 23.00 | 29.00 | 56.85 | 58.06 | 23.08 | 72.39 | 53.57 | 35.19 | 21.95 | 45.57 |
| | w/o thinking | 66.16 | 38.34 | 40.00 | 26.50 | 29.65 | 56.85 | 57.53 | 25.00 | 81.34 | 40.82 | 35.19 | 36.59 | 51.47 |

## D.4 FUSION BOTTLENECK OF MULTI-VIEW STRUCTURAL IMAGES

This section presents an ablation study on multi-view image inputs, designed to investigate the impact of using all six protein structure views versus a single view on the model's average accuracy. As shown in Table 7, the experiment indicates that the model's performance decreases when all six views are provided as input.

Table 7: The Impact of Different Input Views on the Performance of the Qwen2.5-VL-32B. Performance is measured by accuracy (%).

| Model | CA | BP | MF | EC | PW | AS | CF | MP | TM | CC | PH | TA | AVG |
|---|---|---|---|---|---|---|---|---|---|---|---|---|---|
| All | 31.50 | 25.39 | 27.18 | 22.50 | 33.50 | 41.78 | 29.57 | 25.00 | 44.03 | 47.96 | 42.59 | 29.27 | 33.05 |
| Back | 30.50 | 23.83 | 26.67 | 24.50 | 28.00 | 30.14 | 32.80 | 21.15 | 41.79 | 51.53 | 48.15 | 24.39 | 31.88 |
| Front | 35.00 | 26.42 | 29.74 | 21.00 | 27.50 | 30.14 | 37.10 | 21.15 | 51.49 | 50.51 | 51.85 | 19.51 | 33.61 |
| Bottom | 29.50 | 29.53 | 33.85 | 24.00 | 29.50 | 33.56 | 34.41 | 26.92 | 44.03 | 51.53 | 44.44 | 21.95 | 33.88 |
| Top | 30.00 | 25.39 | 29.74 | 25.50 | 24.50 | 34.93 | 34.41 | 23.08 | 45.52 | 48.47 | 48.15 | 24.39 | 32.61 |
| Left | 25.50 | 25.39 | 35.38 | 30.00 | 28.00 | 36.30 | 34.95 | 21.15 | 50.75 | 49.49 | 50.00 | 19.51 | 34.17 |
| Right | 26.50 | 23.83 | 28.21 | 23.00 | 27.50 | 30.82 | 34.95 | 21.15 | 46.27 | 49.49 | 50.00 | 21.95 | 31.77 |

Table 8: Performance comparison of different structural input modalities.

| Model | CA | BP | MF | EC | PW | AS | CF | MP | TM | CC | PH | TA | AVG |
|---|---|---|---|---|---|---|---|---|---|---|---|---|---|
| projection images | | | | | | | | | | | | | |
| GPT-5 | 79.50 | 68.91 | 75.90 | 24.00 | 30.50 | 84.93 | 57.53 | 61.54 | 75.37 | 73.98 | 46.30 | 17.07 | 60.65 |
| GPT-o3 | 72.00 | 60.62 | 70.77 | 24.50 | 21.00 | 81.51 | 59.14 | 63.46 | 79.10 | 67.35 | 44.44 | 17.07 | 56.48 |
| Gemini-2.5-pro | 55.00 | 57.51 | 57.44 | 27.50 | 16.00 | 46.58 | 53.76 | 30.77 | 73.13 | 63.78 | 42.59 | 29.27 | 47.97 |
| Claude-3.7-sonnet | 66.50 | 42.49 | 52.82 | 23.00 | 29.00 | 56.85 | 58.06 | 23.08 | 72.39 | 53.57 | 35.19 | 21.95 | 47.58 |
| Gemini-2.0-flash | 40.00 | 32.12 | 44.62 | 24.00 | 29.50 | 33.56 | 39.78 | 28.85 | 72.39 | 58.16 | 29.63 | 36.59 | 39.84 |
| GPT-4o | 38.50 | 38.86 | 41.54 | 27.00 | 24.50 | 36.99 | 30.11 | 28.85 | 53.73 | 46.94 | 44.44 | 14.63 | 36.45 |
| Qwen2.5-VL-32B | 29.50 | 24.35 | 24.62 | 24.00 | 27.64 | 34.48 | 39.33 | 21.15 | 52.24 | 33.67 | 42.59 | 14.63 | 30.98 |
| InternVL3-78B | 30.00 | 25.91 | 26.15 | 24.50 | 19.00 | 24.66 | 33.87 | 21.15 | 32.09 | 22.96 | 27.78 | 19.51 | 26.10 |
| 3Di | | | | | | | | | | | | | |
| GPT-5 | 72.00 | 69.43 | 67.69 | 27.00 | 27.50 | 80.82 | 52.69 | 57.69 | 70.15 | 64.29 | 40.74 | 19.51 | 56.48 |
| o3 | 70.69 | 59.59 | 68.05 | 25.50 | 28.50 | 79.45 | 51.08 | 63.46 | 72.18 | 63.78 | 42.59 | 26.83 | 53.42 |
| Gemini-2.5-pro | 52.02 | 49.47 | 47.42 | 26.50 | 19.89 | 47.59 | 41.34 | 34.62 | 51.61 | 55.50 | 37.04 | 19.51 | 40.01 |
| Claude-3.7-sonnet | 28.50 | 26.94 | 26.15 | 27.50 | 17.00 | 37.67 | 30.11 | 21.15 | 43.28 | 35.71 | 38.89 | 21.95 | 29.44 |
| Gemini-2.0-flash | 33.50 | 34.20 | 43.08 | 27.50 | 43.50 | 30.14 | 37.63 | 28.85 | 56.72 | 56.12 | 44.44 | 26.83 | 39.45 |
| GPT-4o | 32.00 | 32.64 | 44.10 | 26.50 | 21.00 | 30.82 | 34.95 | 9.62 | 46.27 | 47.45 | 31.48 | 24.39 | 33.67 |
| Qwen2.5-VL-32B | 28.00 | 24.87 | 29.74 | 23.00 | 31.50 | 32.19 | 26.88 | 23.08 | 50.75 | 40.31 | 38.89 | 21.95 | 31.00 |
| InternVL3-78B | 27.50 | 2746 | 27.69 | 23.00 | 26.50 | 21.92 | 32.26 | 23.08 | 42.54 | 30.61 | 42.59 | 24.39 | 28.66 |

## D.5 ANALYSIS OF DIFFERENT STRUCTURAL INPUT MODALITIES

This section presents an ablation study on alternative structural encodings, designed to investigate the impact of using 3Di structural sequences Van Kempen et al. (2024) (a text-based 3D representation) versus 2D structural projection images on the model's average accuracy.

As shown in Table 8, the experiment yields a counter-intuitive but insightful finding: Visual projection images generally outperform text-based 3Di sequences for leading general-purpose models. Specifically, for top-tier models such as GPT-5, GPT-o3, and Claude-3.7-sonnet, switching from

projection images to 3Di sequences results in a significant performance drop. For instance, GPT-5's average accuracy decreases from 60.65% to 56.48%, and Claude-3.7-sonnet suffers a drastic decline from 47.58% to 29.44%. This suggests that while these models possess powerful visual encoders capable of interpreting spatial features from 2D projections, they struggle to interpret the specialized grammar of the 3Di alphabet, which is likely absent from their general pre-training corpora. Without specific fine-tuning, the 3Di sequence acts as noise rather than a semantic signal.

## D.6 IMPACT OF TASK-SPECIFIC ADAPTATION ON SPECIALIZED MODELS

To explicitly verify whether the zero-shot performance deficit of specialized models (SLLMs) stems from a lack of domain knowledge or a failure in instruction alignment, we conducted task-specific adaptation and evaluated the changes in both Accuracy (Table 9) and Pass Rate (Table 10).For each task, we selected a single high-quality response from the 2020 dataset to serve as the one-shot exemplar.

Table 9: Comparison of SLLMs performance before and after task-specific adaptation.

| Model | CA | BP | MF | EC | PW | AS | CF | MP | TM | CC | PH | TA | AVG |
|---|---|---|---|---|---|---|---|---|---|---|---|---|---|
| w/o task-specific adaptation | | | | | | | | | | | | | |
| EvoLlama | 25.50 | 25.91 | 22.05 | 27.50 | 14.50 | 24.66 | 29.57 | 23.08 | 20.15 | 27.55 | 27.78 | 21.95 | 24.26 |
| Evolla | 24.00 | 27.98 | 24.10 | 17.00 | 21.00 | 30.82 | 23.12 | 25.00 | 24.63 | 15.31 | 25.93 | 26.83 | 22.98 |
| w task-specific adaptation | | | | | | | | | | | | | |
| EvoLlama | 33.00 | 32.12 | 24.10 | 27.00 | 19.00 | 25.34 | 35.48 | 19.23 | 22.39 | 50.00 | 31.48 | 34.15 | 30.00 |
| Evolla | 26.50 | 25.39 | 23.08 | 23.50 | 25.50 | 19.18 | 26.34 | 19.23 | 20.90 | 15.31 | 29.63 | 26.83 | 23.21 |

Table 10: Comparison of pass rates for SLLMs before and after task-specific adaptation.

| Model | CA | BP | MF | EC | PW | AS | CF | MP | TM | CC | PH | TA |
|---|---|---|---|---|---|---|---|---|---|---|---|---|
| w/o task-specific adaptation | | | | | | | | | | | | |
| EvoLlama | 40.00 | 85.00 | 73.33 | 45.50 | 71.00 | 63.70 | 80.10 | 80.80 | 46.30 | 58.70 | 63.00 | 46.30 |
| Evolla | 0.00 | 0.00 | 0.00 | 0.00 | 0.00 | 0.00 | 0.00 | 0.00 | 0.00 | 0.00 | 0.00 | 0.00 |
| w task-specific adaptation | | | | | | | | | | | | |
| EvoLlama | 100.00 | 100.00 | 100.00 | 100.00 | 98.50 | 100.00 | 100.00 | 100.00 | 98.51 | 99.49 | 100.00 | 100.00 |
| Evolla | 35.50 | 9.84 | 16.92 | 14.50 | 10.50 | 4.11 | 11.29 | 7.69 | 21.64 | 1.53 | 11.11 | 7.32 |

The tables indicate that one-shot learninig enhances model performance. For EvoLlama, this resulted in a significant boost in average accuracy (24.26% to 30.00%) and a near-perfect instruction adherence rate across all tasks. Conversely, Evolla benefited minimally from alignment, exhibiting only a marginal rise in accuracy (22.98% to 23.21%) while maintaining low pass rates. In summary, our findings underscore that zero-shot evaluation alone may underestimate the capabilities of specialized models due to their varying degrees of instruction-following proficiency. While task-specific adaptation can successfully unlock the latent knowledge of capable models like EvoLlama, it is not a panacea for all architectures.

## D.7 SUPPLEMENTARY EVALUATION ON PROTEIN INTERACTIONS

Table 11 presents the evaluation results on the newly added CPI task, contrasting them with the average performance on the 12 core single-protein tasks, this task consists of 195 questions. The comparison reveals a distinct system-level performance gap, highlighting the increased complexity of modeling biological interactions versus intrinsic properties. As shown in Table 11, almost all models exhibit a sharp decline in accuracy when tasked with predicting interactions. For instance, Claude-3.7-sonnet suffers the most dramatic drop, with accuracy falling from 47.58% to 25.13% ($\Delta$ -22.45). Similarly, top-tier models like GPT-5 and Gemini-2.5-pro experience profound regressions of $-18.60\%$ and $-18.74\%$, respectively. This indicates that while these models have acquired substantial knowledge about individual protein characteristics, they struggle to reason about the dynamic and conditional nature of molecular recognition between proteins and small molecules.

Table 11: Performance comparison between single-protein intrinsic properties and CPI.

| Model | Single-Protein Avg (12 Tasks) | CPI | $\Delta$ (Gap) |
|---|---|---|---|
| Qwen3-32B | 40.23 | 24.62 | -15.61 |
| Deepseek-v3 | 38.95 | 30.77 | -8.18 |
| Deepseek-r1 | 38.62 | 22.56 | -16.06 |
| Llama3.3-70B | 37.67 | 23.59 | -14.08 |
| Qwen2.5-72B | 35.98 | 31.28 | -4.70 |
| Qwen2.5-32B | 35.28 | 29.23 | -6.05 |
| GPT-5 | 60.65 | 42.05 | -18.60 |
| o3 | 56.48 | 37.95 | -18.53 |
| Gemini-2.5-pro | 47.97 | 29.23 | -18.74 |
| Claude-3.7-sonnet | 47.58 | 25.13 | -22.45 |
| Gemini-2.0-flash | 39.84 | 44.61 | 4.77 |
| GPT-4o | 36.45 | 23.59 | -12.86 |
| Qwen2.5-VL-32B | 30.98 | 24.10 | -6.88 |
| InternVL3-78B | 26.10 | 16.92 | -9.18 |
| BioMedGPT-R1 | 31.83 | 13.33 | -18.50 |
| EvoLlama | 24.26 | 20.51 | -3.75 |
| Evolla | 22.98 | 22.05 | -0.93 |

# E  QUALITATIVE ANALYSIS OF STRUCTURE INPUT

To better understand the phenomenon observed in Figure 4 (A), where the addition of structural images often interferes with model performance, we conducted a qualitative analysis of model outputs. We compared the Chain-of-Thought (CoT) reasoning processes of models under two settings: Sequence-Only versus Sequence + Structure.

**Case Study 1**

**Task:** EC
**Model:** Claude-3.7-sonnet
**Target Protein:** A0A9E7S5B9
**Key Observation:** The model correctly identified the functional motif from the sequence but was misled by visual artifacts in the 2D structure projection. The addition of structural information caused the model to choose the wrong answer.

| Modality | Generated Reasoning (Excerpt) | Prediction |
|---|---|---|
| Sequence-Only | ... The sequence contains key motifs typical of aldo-keto reductases (AKRs), including: The presence of "GFGT" motif near the N-terminus. The "LFITS" sequence which is part of a conserved region in AKRs. References to "DLYL" which appears in many AKR family proteins. The overall sequence length ($\tilde{3}00$ amino acids) is consistent with AKRs. Looking at specific clues in the sequence: "GFGTCIPDP" is a signature pattern found in aldo-keto reductases. "LDLYLIHM-PLR" is another conserved region in AKRs. The presence of NAD(P)H binding sites indicated by "DTASCY" ... The specific number 1.1.1.440 represents a particular aldo-keto reductase, and the sequence features align with this classification. | Correct |
| Sequence + Structure | ... From the protein images, I can see: A well-folded globular protein with mixed $\alpha$-helices and $\beta$-sheets. The structure appears to have multiple domains. The folding pattern suggests an enzyme with a defined active site. The overall architecture is consistent with a kinase-like fold. ... The protein sequence and structure are most consistent with a protein kinase (2.7.11.24), specifically a MAP kinase kinase kinase (MAPKKK). | Incorrect |

**Case Study 2**

**Task:** CC
**Model:** Claude-3.7-sonnet
**Target Protein:** Q8U3G4
**Key Observation:** The model correctly identified the functional motif from the sequence but was misled by visual artifacts in the 2D structure projection. The addition of structural information caused the model to choose the wrong answer.

| Modality | Generated Reasoning (Excerpt) | Prediction |
|---|---|---|
| Sequence-Only | ... Looking at the sequence, I notice several key features: The presence of "GRGKS" motif (around position 270) which resembles a Walker A motif (P-loop) often found in ATP/GTP-binding proteins Terms like "DEAD" or "DEAA" (I see "DEAA" around position 343) are often associated with RNA helicases The sequence contains "GRGKSVSI-GIASVGLA" which strongly suggests a nucleotide-binding domain Mentions of "ATP" and "ADP" in the motifs suggest this enzyme uses ATP hydrolysis. The protein appears to have: Nucleotide binding domains ATPase activity Helicase-like features ... Among the choices, option A involves ATP hydrolysis and tRNA modification, which is consistent with the sequence features I identified. The protein appears to have RNA-interacting capabilities and ATP utilization, which would be necessary for the tRNA modification reaction described in option A. | Correct |
| Sequence + Structure | ... Key features I notice in the sequence and structure: The protein is large and multi-domain, with a complex tertiary structure. I can see multiple alpha helices and beta sheets arranged in a pattern typical of nucleotide-binding enzymes. The sequence contains motifs like "GRGKS" which is reminiscent of a P-loop/Walker A motif found in ATP-binding proteins. Terms like "ATP", "ADP", "phosphate" appear in some of the reaction options. Looking more closely at sequence motifs: "GRGKSVSIGIASVG" appears to be a nucleotide-binding motif "VITDNGLIKEVYAK" suggests a catalytic domain. The presence of "VVRIATHPDA" suggests a phosphorylation-related activity. The structure shows a large multi-domain protein with distinct regions that could accommodate nucleotide binding and substrate binding. The rainbow coloring shows what appears to be multiple domains arranged in a way consistent with kinase or phosphotransferase activity. ... Option C describes a kinase reaction: L-aspartate + ATP = 4-phospho-L-aspartate + ADP. This matches well with the protein's features: The nucleotide-binding motifs would accommodate ATP. The structure has domains consistent with kinase activity. The reaction is a simple phosphoryl transfer from ATP to aspartate. This protein appears to be aspartate kinase, which catalyzes the first step in the biosynthesis of several amino acids by phosphorylating aspartate. | Incorrect |

**Case Study 3**

**Task:** EC
**Model:** o3
**Target Protein:** I0AIT9
**Key Observation:** The sequence analysis yielded ambiguous signals regarding the protein's specific category, resulting in an incorrect initial prediction. However, the 2D structure projections provided decisive visual evidence of global topological features that were not apparent from the sequence alone. The addition of structural information disambiguated the conflicting signals, enabling the model to correct its reasoning and choose the right answer.

| Modality | Generated Reasoning (Excerpt) | Prediction |
|---|---|---|
| Sequence-Only | ... The protein sequence is 780 amino-acids long and, apart from a few short stretches, it is predominantly hydrophilic. There is no obvious N-terminal signal peptide or extended hydrophobic segment that would anchor it in a membrane. Consequently, it is most likely a soluble, cytosolic enzyme... Option B (EC 2.4.1.391) designates a glycosyltransferase. Many bacterial inverting or retaining glycosyltransferases involved in polysaccharide, glycolipid or glycoprotein synthesis are large, soluble proteins of 600–800 aa that lack obvious signal peptides. Their sequences characteristically contain several "DXH"/"H(R/K)D" and "E/D–E" acidic motifs for nucleotide-sugar binding and catalysis, which are present multiple times in the given protein. The overall size and motif pattern therefore fit a glycosyltransferase far better than any of the other enzyme classes offered. Because only option B is compatible with the length, sub-cellular localization, and conserved sequence motifs, it is the most plausible choice. | Incorrect |
| Sequence + Structure | ... Size and fold. The polypeptide comprises 840 amino-acid residues and the structure shown is a largely $\alpha/\beta$ cytosolic dimer. It is far larger than typical $\beta$-lactamases (260 aa, option D) or endo-xylanases (200 aa, option A), and it lacks the N-terminal signal peptide or obvious trans-membrane segments expected for most secreted glycosidases or glycosyltransferases (option B)... Therefore the only option consistent with the sequence and structural features is EC 2.8.1.7. | Correct |

# F   RESPONSE ERROR ANALYSIS

A significant finding in our experiments is that despite being fine-tuned with domain knowledge, small, domain-specific models (SLLMs) generally underperform large, general-purpose models. This section aims to analyze a key reason for this performance gap: the difference in instruction-following ability. During our testing, we observed that SLLMs more frequently fail to generate valid or correctly formatted answers. To quantify this issue, we introduce the "Pass Rate" as an evaluation metric. This metric measures the proportion of answers a model can generate that strictly adhere to the format specified in our prompt ("1. reasoning: [...]; 2. answer: [A/B/C/D]"), regardless of whether the content of the answer is correct. This metric directly reflects the model's ability to understand and execute complex instructions.

As shown in Table 12, the analysis reveals a significant gap between general-purpose and specialized models in this capability. Top-tier general-purpose models (like GPT-5 and Gemini-2.5-pro) have a Pass Rate approaching $100\%$. In contrast, the performance of SLLMs is much poorer. For example, according to our statistics, the pass rates of Evolla and BioMedGPT-R1 are close to $0\%$.

To provide a more detailed, qualitative view of the instruction-following failures discussed in the Error Analysis section, Table 13 presents representative examples of incorrectly formatted responses generated by SLLMs. The responses from BioMedGPT-R1 exemplify a common failure mode where the model adopts a conversational. Despite identifying the task, the model fails to struc-

Table 12: Comparison of model pass rates

| Model | CA | BP | MF | EC | PW | AS | CF | MP | TM | CC | PH | TA |
|---|---|---|---|---|---|---|---|---|---|---|---|---|
| Qwen3-32B | 64.00 | 71.00 | 62.10 | 89.00 | 74.50 | 84.90 | 73.70 | 84.60 | 89.60 | 91.60 | 79.60 | 70.70 |
| Deepseek-v3 | 100.00 | 100.00 | 100.00 | 100.00 | 99.50 | 100.00 | 99.46 | 100.00 | 100.00 | 98.93 | 100.00 | 100.00 |
| Qwen2.5-32B | 55.00 | 84.46 | 96.41 | 67.00 | 68.50 | 88.36 | 40.86 | 73.08 | 84.33 | 89.29 | 94.44 | 65.85 |
| GPT-5 | 100.00 | 100.00 | 100.00 | 99.50 | 100.00 | 100.00 | 100.00 | 96.15 | 99.25 | 100.00 | 100.00 | 100.00 |
| GPT-o3 | 100.00 | 100.00 | 100.00 | 99.50 | 100.00 | 100.00 | 99.46 | 100.00 | 99.25 | 100.00 | 100.00 | 100.00 |
| Gemini-2.5-pro | 95.00 | 94.30 | 93.80 | 93.00 | 94.50 | 71.20 | 100.00 | 84.60 | 69.40 | 89.80 | 98.10 | 90.20 |
| Claude-3.7-sonnet | 94.00 | 41.45 | 50.80 | 96.50 | 82.50 | 97.26 | 97.85 | 86.54 | 89.55 | 45.92 | 96.30 | 100.00 |
| Gemini-2.5-pro | 99.50 | 100.00 | 100.00 | 100.00 | 100.00 | 100.00 | 100.00 | 98.08 | 100.00 | 100.00 | 100.00 | 100.00 |
| GPT-4o | 98.50 | 76.20 | 70.30 | 98.50 | 96.00 | 91.80 | 96.20 | 94.20 | 95.50 | 74.00 | 90.70 | 80.50 |
| Qwen2.5-VL-32B | 67.50 | 32.10 | 39.00 | 83.50 | 96.50 | 100.00 | 33.33 | 100.00 | 100.00 | 92.30 | 100.00 | 95.10 |
| InternVL3-78B | 0.00 | 0.50 | 0.00 | 1.00 | 2.00 | 0.70 | 0.00 | 0.00 | 3.00 | 0.00 | 1.90 | 0.00 |
| BioMedGPT-R1 | 0.00 | 0.00 | 0.00 | 0.00 | 0.00 | 0.00 | 0.00 | 0.00 | 0.00 | 0.00 | 0.00 | 0.00 |
| EvoLlama | 40.00 | 85.00 | 73.33 | 45.50 | 71.00 | 63.70 | 80.10 | 80.80 | 46.30 | 58.70 | 63.00 | 46.30 |
| Evolla | 0.00 | 0.00 | 0.00 | 0.00 | 0.00 | 0.00 | 0.00 | 0.00 | 0.00 | 0.00 | 0.00 | 0.00 |

ture its output according to the required reasoning and answer format, instead providing a narrative followed by an incorrectly formatted final choice. The case of Evolla highlights a different type of error: the model successfully generates a structured, step-by-step reasoning process, but it uses its own self-devised format rather than the one specified in the prompt. These cases demonstrate that the low Pass Rates of SLLMs are not due to a lack of response, but rather a fundamental inability to adhere to complex formatting constraints. This underscores our conclusion that a significant performance bottleneck for these specialized models is their limited instruction-following capability, a foundational skill where larger general-purpose models excel.

## G  TASKS DEFINITION

**Catalytic Activity** This task aims at predicting the specific chemical reaction catalyzed by a given protein. We use experimentally validated catalytic activity annotations from the UniProt database as the ground truth labels to ensure the reliability of the evaluation. Accurately predicting catalytic activity is fundamental to deciphering a protein's role in complex metabolic pathways and is key to discovering new enzymes. Furthermore, this capability has profound applicational value in synthetic biology and industrial biocatalysis, where it is central to designing and optimizing biomanufacturing processes.

**Enzyme Commission Number** This task predicts enzyme function by assigning the corresponding EC number to a protein sequence. We employ the experimentally verified EC annotations from the UniProt database as the gold standard. The EC number is a four-level hierarchical classification system that provides a standardized language for enzyme function, and its accurate prediction is crucial for systematically understanding enzymatic reactions and discovering novel enzyme activities. This technology has wide-ranging applications in biotechnology, including the optimization of industrial enzymes and guidance for drug development targeting specific enzyme classes.

**Molecular Function** This task based on the Gene Ontology (GO) framework, designed to predict the specific activities a protein performs at the molecular level, such as "ATP binding" or "transporter activity". Our ground truth labels are derived from annotations in the UniProt-GOA database that are supported by experimental evidence, ensuring the highest quality of annotation. This task directly probes the model's understanding of a protein's intrinsic capabilities, independent of its cellular location or the biological processes it participates in. Accurate MF prediction is a cornerstone of functional genomics, essential for annotating newly discovered genes and identifying potential drug targets.

**Biological Process** This task based on the Gene Ontology (GO) framework, which aims to identify the larger-scale biological processes a protein is involved in, such as "glycolysis" or "cell signaling". Unlike MF, which focuses on a single molecular activity, BP places a protein's function into a broader physiological context, connecting molecular events to the overall operation of the cell. This task evaluates a model's higher-level understanding of the roles proteins play within complex living systems. A deep understanding of the biological processes proteins participate in is vital for uncovering disease pathogenesis and discovering new therapeutic strategies.

**Pathway** This task requires the model to assign a given protein to its specific metabolic or signaling pathway. We compiled validated pathway annotation information from databases such as UniProt to serve as the ground truth. The accurate identification of a protein's pathway is critical for systems biology research, as it helps to elucidate complex biological regulatory networks and understand how cellular processes are coordinately regulated. This knowledge is valuable for understanding how drugs affect an entire cellular system by targeting a single protein and for applications in metabolic engineering.

**Active Site** This task requires the model to identify key amino acid residues in a protein sequence that are directly involved in substrate binding or catalysis. The precise prediction of active sites is central to structural biology and computational drug design, as these residues are the primary targets for developing inhibitors or activators. These sites are often evolutionarily conserved and possess a unique chemical environment tailored for a specific reaction. Therefore, a model's ability to identify them from sequence alone is a strong indicator of its understanding of the structure-function relationship.

**Cofactor** This task aims to predict the non-protein chemical components a protein must bind to perform its biological function. These include metal ions (e.g., zinc, magnesium) or organic molecules (e.g., NAD+, FAD). These cofactors often act as "helper molecules" that are directly involved in catalysis or are essential for maintaining the protein's structural integrity. Understanding a protein's cofactor requirements is crucial for successful enzyme engineering, nutritional science, and the diagnosis and treatment of diseases related to cofactor metabolism deficiencies.

**Motif Position** This task requires the model to pinpoint the exact location of conserved, short sequence patterns (motifs) with specific biological significance within a protein sequence. These motifs often serve as functional "signatures", such as the "zinc finger" motif associated with DNA binding or specific phosphorylation sites. Because these patterns are evolutionarily conserved and directly linked to functions like binding, catalysis, or post-translational modification, accurately identifying their positions is an effective means of rapidly inferring the function of newly discovered proteins.

**Transmembrane** This task is designed to determine whether a protein is embedded within a cell membrane. Transmembrane proteins typically contain one or more hydrophobic alpha-helical segments that span the lipid bilayer, functioning as channels, transporters, or receptors. They act as the "gatekeepers" of the cell, playing a central role in signal transduction, substance transport, and cell-to-cell communication. Given that they constitute a large fraction of all known drug targets, their accurate identification is of extremely high value for drug discovery and fundamental cell biology research.

**Cellular Component** This task is based on the Gene Ontology (GO) framework, which aims to determine a protein's specific subcellular localization, such as the "nucleus", "mitochondrion", or "cytoplasm". A protein's location is intrinsically linked to its function, as it dictates its local environment and available potential interaction partners. For example, a protein located in the nucleus is likely involved in DNA replication or transcription. Therefore, the accurate prediction of cellular components is fundamental to understanding the division of labor among proteins to carry out complex operations within the cell.

**Optimal pH** This task requires the model to predict the environmental pH at which a protein exhibits its maximum biological activity. The ambient pH directly influences the ionization state of amino acid side chains, which in turn affects the protein's three-dimensional structure and the chemical properties of its active site. This prediction is crucial for industrial biotechnology, as it determines an enzyme's efficiency and stability in specific production processes. It also helps in understanding how proteins adapt their function in cellular compartments with different acid-base environments, such as the acidic lysosome.

**Thermal Adaptation** This task aims to infer the optimal growth temperature category (e.g., psychrophilic, mesophilic, or thermophilic) of the organism from which a protein originates. This capability has immense potential for discovering and engineering stable enzymes that remain active under extreme industrial conditions, such as in high-temperature detergents or low-temperature food processing.

## H    GENERAL PROMPT

In this section, we present the general prompt used for question-answering.

---

**System Message**

You are an excellent scientist. Your should analyze the provided protein-related task carefully and choose the correct answer form the multiple-choice options.
The current task is about {task name}, which {task description}. The inputs provided by the user for this task include:
* Protein Sequence: The amino acid sequence of the protein.
* Protein Image: Multiple views of the protein structure.
* Multiple Choices: Options for the answer.
Please think step-by-step about this problem:
1. Analyze the protein sequence and structure carefully
2. Consider the biological context and function
3. Evaluate each multiple choice option
4. Provide your reasoning process
5. Finally, give your answer
Provide your response in following format:
1. reasoning: [Your detailed reasoning here]
2. answer: Your final answer, which should be A, B, C or D.

---

## I    EXAMPLES OF QUESTIONS AND ANSWERS

In this section, we show several representative examples of questions and answers for each task.

---

**User Message**

task name: catalytic activity
task description: is to identify the specific chemical reaction catalyzed by a given enzyme. This involves understanding the enzyme's function based on its amino acid sequence and then selecting the correct transformation of substrates into products from a list of possible reactions.
[Protein Sequence]
MAIKLIVGLANPGAEYAATRHNAGAWYVDLLAERLRAPLREEPKFFGYTSRITLEGE
DVRLLVPTTFMNLSGKAVGAMASFYRIQPDEILVAHDELDLPPGVAKFKLGGGHGG
HNGLKDIISKLGNNPNFHRLRVGIGHPGDKNKVVGFVLGKPPVSEQKLIDEAIDEA
ARCTELWFKEGLAKATSRLHTFKAQ
[Protein Image]
back.png,
front.png,
bottom.png,
top.png,
left.png,
right.png
[Multiple Choices]
A: (13S)-hydroperoxy-(9Z,11E,15Z)-octadecatrienoate = etherolenate + H2O
B: 4-nitrobenzoate + 2 NADPH + 2 H(+) = 4-hydroxylaminobenzoate + 2 NADP(+) + H2O
C: an N-acyl-L-alpha-aminoacyl-tRNA + H2O = an N-acyl-L-amino acid + a tRNA + H(+)
D: (2R)-2,3-dihydroxy-3-methylbutanoate = 3-methyl-2-oxobutanoate + H2O

---

**Answer**

1. reasoning: - The protein is small ( 170 aa), soluble, and displays a mixed alpha/beta fold, consistent with peptidyl-tRNA hydrolase (Pth) family proteins, which are typically  170–200 aa monomeric cytosolic enzymes.- The sequence contains hallmark Pth motifs: a catalytic His near the N-terminus and a highly conserved HGGH-containing segment. Notably, the sequence includes "...KFKLGGGHGGHNGLK...", matching the conserved GGH/HGGH motif characteristic of Pth active-site regions.- No signatures of other enzyme classes are present: - A: Allene oxide synthase (a CYP74 P450) would be a  500 aa heme-thiolate enzyme with a Cys heme-binding motif and often a membrane helix—absent here. - B: FMN-dependent nitroreductases are  200+ aa with characteristic FMN/NAD(P)H-binding motifs (e.g., TGW/TGGXGGXG) not observed in this sequence. - D: Dihydroxy-acid dehydratase (IlvD) is a large ( 540 aa) [4Fe–4S] enzyme with Cys-Thr-Cys motifs—absent and size is inconsistent.- The reaction in C precisely matches Pth activity: hydrolysis of peptidyl-/N-acyl-aminoacyl-tRNA to release the N-acyl-amino acid and free tRNA.2. answer: C",

Table 13: Case studies of instruction-following failures in specialized models.

| Model | Response |
|-------|----------|
| **BioMedGPT-R1** | Okay, I'm trying to figure out the cofactor of this protein based on the following information. Hmm, that looks a bit complicated, but I can try to break it down. First, I'm thinking about the function of the provided protein. Involved in the biosynthesis of ent-copalyl diphosphate (ent-CDP) from isopentenyl diphosphate (IPP). Catalyzes the oxidative cyclization of geranyl diphosphate (GPP) to ent-copalyl diphosphate (ent-CDP). Can also act on geranylgeranyl diphosphate (GGPP) as substrate producing ent-copalyl diphosphate (ent-CDP). Then there's a list of possible cofactors: A. pantetheine 4'-phosphate, B. Fe(2+), C. Ca(2+), D. heme. I'm trying to figure out the cofactor of this protein based on the following information. A. pantetheine 4'-phosphate B. Fe(2+) C. Ca(2+) D. heme The cofactor of ent-copalyl diphosphate synthase. |
| | Okay, I'm trying to figure out the cofactor of this protein based on the following information. Hmm, that looks a bit complicated, but I can try to break it down. First, I'm thinking about the function mentioned. Involved in the biosynthesis of ergosterol. Catalyzes the NADPH-dependent 3-hydroxylation of 24,25-dihydrosqualene to 24-hydroxysqualene, the first hydroxylation step in the pathway leading to ergosterol. Then there's the cofactor mentioned. A: FAD; B: Mg(2+); C: heme; D: pantetheine 4'-phosphate. The answer is A. </think >A: FAD; B: Mg(2+); C: heme; D: pantetheine 4'-phosphate. |
| **Evolla** | I'd be happy to help with the Enzyme Commission (EC) Number Prediction task.**Step 1: Analyze the protein sequence and structure**The provided protein sequence is a long chain of amino acids, which can be analyzed for patterns, motifs, and functional domains. However, without the 3D structure, it's challenging to identify specific structural features that might aid in EC number prediction. Since the structure is not provided as a text format, I will focus on the sequence analysis.**Step 2: Consider the biological context and function**The protein sequence does not contain any obvious functional domains or motifs that are typically associated with specific EC numbers. However, the sequence does contain a mix of hydrophobic and hydrophilic amino acids, which suggests that the protein may be involved in a biological process that requires interactions with other molecules.**Step 3: Evaluate each multiple choice option**A. 1.14.15.37: This EC number corresponds to a family of enzymes that catalyze the oxidation of a sulfur-containing compound, which is not immediately apparent from the provided sequence.B. 4.4.1.11: This EC number corresponds to a family of enzymes that catalyze the hydrolysis of a peptide bond, which is not a clear match based on the sequence.C. 2.4.2.14: This EC number corresponds to a family of enzymes that catalyze the transfer of a glycosyl group, which is not a clear match based on the sequence.D. 4.3.1.12: This EC number corresponds to a family of enzymes that catalyze the transfer of a phosphorus-containing group, which is not a clear match based on the sequence.**Step 4: Provide my reasoning process**Based on the sequence analysis, I did not find any obvious functional domains or motifs that are typically associated with specific EC numbers. The sequence does not contain any obvious patterns or features that would suggest a specific EC number. Therefore, I will rely on the multiple choice options to make an educated guess.**Step 5: Provide my answer**1. reasoning: The sequence analysis did not reveal any clear functional domains or motifs that are typically associated with specific EC numbers. The multiple choice options do not provide a clear match based on the sequence.2. answer: I will choose option D. 4.3.1.12 as the most plausible answer, although it is not a clear match based on the sequence. |

