# OpenReview forum: "LiveProteinBench: A Contamination-Free Benchmark for Assessing Models' Specialized Capabilities in Protein Science"
_ICLR.cc/2026/Conference — Submitted to ICLR 2026_

### Official Review · Reviewer_BW9u · 2025-10-31

**Soundness:** 2
**Presentation:** 1
**Contribution:** 2
**Rating:** 2
**Confidence:** 4

**Summary:**

This manuscript presents LiveProteinBench, a benchmark designed to evaluate large language models (LLMs) on protein science tasks using a strictly contamination-free framework. It includes 12 biologically diverse tasks—ranging from function prediction to structural reasoning—based on proteins validated after January 1, 2025, ensuring no overlap with pretraining data. However, the paper suffers from key weaknesses, including limited methodological innovation, poor writing quality, and a lack of clarity regarding the dataset.

**Strengths:**

1-	The use of post-2025 protein entries ensures that none of the test data overlaps with pretraining corpora, addressing a critical issue in LLM evaluation: data leakage.
2-	Broad evaluation across general and domain-Specific LLMs.

**Weaknesses:**

1-	Lack of methodological innovation and dataset accessibility. The manuscript does not present any clear methodological innovation beyond the temporal filtering strategy. Furthermore, the authors do not provide access to the benchmark dataset.
2-	Authors didn’t provide details on the proteins analyzed. They just mentioned selection criteria from public database, but any further biological information (e.g., biological diversity, sequence novelty, representative of real challenges, …etc). Therefore, it is challenging to assess the contamination-free claims.
3-	The benchmark’s dataset size is limited, with only ~2,000 proteins. This is substantially smaller than other recent benchmarks such as PROBE and ProteinLMBench, which use tens of thousands of proteins and provide public access to data and evaluation pipelines. The small scale of LiveProteinBench may limit its generalizability and statistical robustness.
4-	Lack of fine-tuning or adaptation experiments especially for small-scale specialized models (SLLMs). Only zero-shot evaluation is reported; no exploration of how models could improve with task-specific training. This is a significant limitation, as recent literature [Schmirler et al. Nature Comm 2024] demonstrated that fine-tuning can substantially improve performance on protein-specific tasks
5-	The current version of the manuscript needs significant improvement in writing quality. The paper is densely written and lacks clear presentation and methodological articulation of key contributions such as multi-modal integration. For example, how were the protein images paired with sequence data? How multi-modal input was provided to the model? etc. In addition to vague sentences such as “Genuinely unlocking the secrets of life requires these models to move beyond merely processing sequence information and to demonstrate multiple advanced capabilities.”

**Questions:**

N/A

---

> ### Author Response · Authors · 2025-11-26
> **Response to Reviewer BW9u**
>
> We thank the reviewer for the constructive comments. Regarding the concerns of the reviewer BW9u, we provide the following responses.
>
> **W1: Lack of methodological innovation and dataset accessibility.**
> - We clarify that this is primarily an evaluation-oriented paper, rather than a study proposing a new model architecture. In this context, our contributions are centered on the rigorous design, construction, and extensive analysis of a benchmark intended to solve a fundamental crisis in current LLM evaluation: data contamination. Specifically, our contributions are followed:
> - We introduce a rigorous "Live Data" construction framework. Unlike traditional benchmarks that rely on static historical data, our approach strictly enforces a temporal split (post-January 1, 2025).
> - The necessity of this design is empirically validated in Figure 3B. Our experiments reveal a massive performance gap between models' performance on "seen" historical data versus "unseen" LiveProteinBench data. This quantitative evidence highlights that previous high scores were largely due to memorization, proving that data contamination is a severe and often overlooked issue that our benchmark successfully exposes.
> - We conducted an extensive evaluation of over 10 representative models across 12 diverse protein tasks. This extensive analysis provides the community with a realistic view of current capabilities.
> - We have provided representative examples of the questions in the Supplementary Material.
>
> **W2: Authors didn’t provide details on the proteins analyzed.**
> - To quantify the sequence novelty of our benchmark, we employed Blastp to search our test proteins against the UniProt reference database. Our dataset spans a broad spectrum of sequence similarities, ranging from highly homologous instances (Identity $\ge$ 90%) to challenging cases with very low similarity (Identity < 30.0%).
>
> **W3: Limited benchmark’s dataset size.**
> - LiveProteinBench primarily focuses on the real-time nature of data rather than its scale. Given the strict temporal window of 'post-January 1, 2025', the pool of newly released, experimentally reviewed proteins is inherently limited. Consequently, amassing tens of thousands of proteins, comparable to the scale of benchmarks like PROBE or ProteinLMBench, is currently infeasible within this rigorous timeframe.
> - Furthermore, it is critical to note that the vast majority of data in these larger benchmarks was released prior to the training cutoffs of current foundation models. This introduces a significant risk of data contamination, making them less suitable for objectively evaluating the true generalization capabilities of LLMs. 、
> - Finally, as detailed in Section 3.2, our automated update pipeline is designed to address this limitation by ensuring the continuous growth of the benchmark's scale as new data becomes available.
>
> **W4:  Impact of instruction fine-tuning on specialized models.**
> - We could not find the paper cited as [Schmirler et al. Nature Comm 2024] and would appreciate it if you could share more specific details, such as the full title.
> - We applied task-specific adaptation to two underperforming SLLMs via one-shot in-context learning. This approach operates as a form of implicit gradient descent, allowing the models to align with the task distribution using task-specific examples (1. reasoning... 2. answer...) within the input context. The results are presented in the table below,
> | Model | CA | BP | MF | EC | PW | AS | CF | MP | TM | CC | PH | TA | AVG |
> | :--- | :---: | :---: | :---: | :---: | :---: | :---: | :---: | :---: | :---: | :---: | :---: | :---: | :---: |
> | **w/o task-specific adaptation** | | | | | | | | | | | | | |
> | EvoLlama | 25.50 | 25.91 | 22.05 | 27.50 | 14.50 | 24.66 | 29.57 | 23.08 | 20.15 | 27.55 | 27.78 | 21.95 | 24.26 |
> | Evolla | 24.00 | 27.98 | 24.10 | 17.00 | 21.00 | 30.82 | 23.12 | 25.00 | 24.63 | 15.31 | 25.93 | 26.83 | 22.98 |
> | **w task-specific adaptation** | | | | | | | | | | | | | |
> | EvoLlama | 33.00 | 32.12 | 24.10 | 27.00 | 19.00 | 25.34 | 35.48 | 19.23 | 22.39 | 50.00 | 31.48 | 34.15 | 30.00 |
> | Evolla | 26.50 | 25.39 | 23.08 | 23.50 | 25.50 | 19.18 | 26.34 | 19.23 | 20.90 | 15.31 | 29.63 | 26.83 | 23.21 |
> - As shown in the table, task-specific adaptation improves the model's performance. We have incorporated these experimental results and the corresponding analysis into the revised manuscript. Please refer to Appendix D.6 for more results and analysis.

---

> ### Author Response · Authors · 2025-11-26
> **Response to Reviewer BW9u**
>
> **W5: The current version of the manuscript needs significant improvement in writing quality.**
> - We will continue to polish the writing of this manuscript. Below is an explanation of how multimodal fusion is performed.
> - For each protein entry, we construct a composite input. The sequence data is embedded directly into the text prompt (as shown in Appendix I), while the structural data consists of 6 orthogonal 2D projection images generated via PyMOL.
> - For MLLMs, we utilize an interleaved input format. The model receives a text prompt containing the amino acid sequence and the specific question first, followed by the 6 images. The model is explicitly instructed in the system prompt to 'analyze the provided protein-related task... based on the protein's sequence and structure'.

---

### Official Review · Reviewer_v6T2 · 2025-11-02

**Soundness:** 3
**Presentation:** 3
**Contribution:** 3
**Rating:** 6
**Confidence:** 4

**Summary:**

The paper introduces LiveProteinBench, a contamination-free and multimodal benchmark for assessing large language models’ (LLMs) capabilities in protein science.

It features 12 structured tasks spanning protein function, structure, and physicochemical property prediction, built exclusively from UniProt entries validated after January 2025 to ensure no data leakage.

The authors benchmark over 10 general-purpose and protein-specific models, revealing that general-purpose models (e.g., GPT-5) outperform specialized ones and that multimodal integration of 3D protein structures remains a significant challenge.

**Strengths:**

The paper is original in proposing a live, contamination-free design for benchmarking LLMs in biology.

The methodology is rigorous, with carefully defined tasks, fair temporal splits, and reproducibility ensured through public databases.

The clarity of the presentation and experimental analyses is high, and the results are significant

**Weaknesses:**

The multimodal evaluation relies on 2D structure projections, which may not fully capture 3D relationships; alternative encodings could be discussed.

The benchmark focuses only on single-protein properties, omitting interactions or dynamics that are crucial in biological contexts.

Evaluation metrics are limited to accuracy.

Limited discussions of related works such as [1, 2, 3]

[1] STELLA: Towards Protein Function Prediction with Multimodal LLMs Integrating Sequence-Structure Representations

[2] Proteingpt: Multimodal llm for protein property prediction and structure understanding

[3] Prot2Text-V2: Protein Function Prediction with Multimodal Contrastive Alignment

**Questions:**

How will LiveProteinBench be maintained to ensure continued contamination-free status as future models update?

Could the authors provide evidence that the cutoff date fully excludes pretraining data from foundation model updates?

---

> ### Author Response · Authors · 2025-11-25
> **Response to Reviewer v6T2**
>
> We thank the reviewer for the constructive comments. Regarding the concerns of the reviewer v6T2, we provide the following responses.
>
> **W1: Analysis of alternative structure encodings.**
> - We thank the reviewer for the valuable feedback. We adopted 3Di [1] as an alternative structural encoding to projection images to benchmark MLLM performance. In this experimental setup, the inputs to the MLLMs consisted of the protein sequence and the 3Di sequence. The performance of MLLMs on a subset of tasks is presented below:
>
> | Model | CA | BP | MF | EC | PW | AS |
> | :--- | :---: | :---: | :---: | :---: | :---: | :---: |
> | GPT-5 | 72.00 | 69.43 | 67.69 | 27.00 | 27.50 | 80.82 |
> | o3 | 70.69 | 59.59 | 68.05 | 25.50 | 28.50 | 79.45 |
> | Gemini-2.5-pro | 52.02 | 49.47 | 47.42 | 26.50 | 19.89 | 47.59 |
> | Claude-3.7-sonnet | 28.50 | 26.94 | 26.15 | 27.50 | 17.00 | 37.67 |
> | Gemini-2.0-flash | 33.50 | 34.20 | 43.08 | 27.50 | 43.50 | 30.14 |
> | GPT-4o | 32.00 | 32.64 | 44.10 | 26.50 | 21.00 | 30.82 |
> | Qwen2.5-VL-32B | 28.00 | 24.87 | 29.74 | 23.00 | 31.50 | 32.19 |
> | InternVL3-78B | 27.50 | 27.46 | 27.69 | 23.00 | 26.50 | 21.92 |
> - The results indicate that visual projection images generally outperform text-based 3Di sequences for leading general-purpose models.
> - Our analysis suggests that these models possess powerful visual encoders capable of interpreting spatial features from 2D projections. They struggle to interpret the specialized grammar of the 3Di alphabet, which is likely absent from their general pre-training corpora.
> - Please refer to Appendix D.5 for more results and analysis.
>
> [1] Michel Van Kempen, Stephanie S Kim, Charlotte Tumescheit, Milot Mirdita, Jeongjae Lee, Cameron LM Gi lchrist, Johannes S¨oding, and Martin Steinegger. Fast and accurate protein structure search with foldseek. Nature biotechnology, 42(2):243–246, 2024.
>
> **W2: Compound-protein interaction results.**
> - We collected experimentally validated compound-protein binding affinity data published in major journals after the start of 2025, sourced from the BindingDB database. From this collection, we randomly selected 195 instances to serve as test questions. The performance of the evaluated models is presented below:
> | Model | Qwen3-32B | Deepseek-v3 | Deepseek-r1 | Llama3.3-70B | Qwen2.5-72B | Qwen2.5-32B | GPT-5 | o3 | Gemini-2.5-pro | Claude-3.7-sonnet | Gemini-2.0-flash | GPT-4o | Qwen2.5-VL-32B | InternVL3-78B | BioMedGPT-R1 | EvoLlama | Evolla |
> | :--- | :---: | :---: | :---: | :---: | :---: | :---: | :---: | :---: | :---: | :---: | :---: | :---: | :---: | :---: | :---: | :---: | :---: |
> | **CPI** | 24.62 | 30.77 | 22.56 | 23.59 | 31.28 | 29.23 | 42.05 | 37.95 | 29.23 | 25.13 | 44.61 | 23.59 | 24.10 | 16.92 | 13.33 | 20.51 | 22.05 |
> - The results indicate that while these models have acquired substantial knowledge about individual protein characteristics, they struggle to reason about the dynamic and conditional nature of molecular recognition between proteins and small molecules.
> - We have incorporated these experimental results and the corresponding analysis into the revised manuscript. Please refer to Appendix D.7 for more results and analysis.
>
> **W3: Evaluation metrics are limited to accuracy.**
> - Since our benchmark is uniformly standardized in a Multiple-Choice Question (MCQ) format, the ground truth is unambiguous. Consequently, accuracy serves as a precise and objective indicator of model performance.
> - Additionally, we introduced the "Pass Rate" metric in Appendix F to evaluate the models' ability to strictly adhere to complex formatting instructions.
> - We did consider evaluating open-ended generation tasks. However, given the high diversity of textual responses and the complexity of protein representations, reliable evaluation would heavily depend on human experts or specialized verifiers, which inevitably introduces additional evaluation error. Therefore, we adopted the MCQ format as a robust first step.
> - Building on our current data, we plan to expand the benchmark in future iterations to include diverse question types and metrics, supported by dedicated judge models to ensure reliable evaluation.
>
> **W4: Limited discussions of related works**
> - We incorporated a discussion of these papers into the 'Related Work' section of the revised manuscript.

---

> ### Author Response · Authors · 2025-11-25
> **Response to Reviewer v6T2**
>
> **Q1: Methods to maintain the contamination-free status of LiveProteinBench for future model updates.**
> - To ensure the long-term contamination-free status of LiveProteinBench, we implement a continuous evaluation protocol, similar to the approach taken by LiveBench and LiveCodeBench.
> - As detailed in Section 3.2, we established an 'automated update pipeline' grounded in these principles. This system is designed to automate data curation by periodically scanning authoritative databases to precisely identify and incorporate the latest protein entries that meet our strict temporal constraints. This guarantees that the benchmark continuously provides high-quality, contamination-free evaluation data.
>
> **Q2: Evidence regarding the cutoff date of LiveProteinBench data excludes pretraining data from foundation model updates.**
> - This can be found in their official technical reports.  For instance, GPT-5 explicitly states that its training data cutoff is September 30, 2024 (https://learn.microsoft.com/en-us/azure/ai-foundry/foundry-models/concepts/models-sold-directly-by-azure?view=foundry-classic&tabs=global-standard-aoai%2Cstandard-chat-completions%2Cglobal-standard&pivots=azure-openai).

---

### Official Review · Reviewer_t8LX · 2025-11-02

**Soundness:** 2
**Presentation:** 3
**Contribution:** 2
**Rating:** 4
**Confidence:** 2

**Summary:**

LiveProteinBench is a contamination-free, multimodal benchmark specifically designed to assess LLM capabilities in protein science. The benchmark includes 12 diverse tasks across functional annotation, structural localization, and physicochemical property prediction, using only post-2025 data from UniProt to ensure no pretraining contamination. The authors evaluate >10 general-purpose and domain-specific models using both sequence-only and sequence+structure modalities.

**Strengths:**

- LiveProteinBench addresses major flaws in existing protein evaluation (contamination, outdated tasks, lack of multimodality) with rigorous dataset construction and “live data” principle.

- The benchmark offers 12 well-structured tasks grounded in validated annotations; task variety enables broad assessment of biological reasoning.

**Weaknesses:**

- The evaluations are zero-shot. It would be valuable to see whether task-tuned or instruction-fine-tuned models can close the generalist-specialist gap.

**Questions:**

- Are there some qualitative examples where structure helped vs. harmed performance to better understand the fusion bottleneck?

---

> ### Author Response · Authors · 2025-11-26
> **Response to Reviewer t8LX**
>
> We thank the reviewer for the constructive comments. Regarding the concerns of the reviewer t8LX, we provide the following responses.
>
> **W1. specialized models.**
> - We applied task-specific adaptation to two underperforming SLLMs via one-shot in-context learning. This approach operates as a form of implicit gradient descent, allowing the models to align with the task distribution using task-specific examples (1. reasoning... 2. answer...) within the input context. The results are presented in the table below,
> | Model | CA | BP | MF | EC | PW | AS | CF | MP | TM | CC | PH | TA | AVG |
> | :--- | :---: | :---: | :---: | :---: | :---: | :---: | :---: | :---: | :---: | :---: | :---: | :---: | :---: |
> | **w/o task-specific adaptation** | | | | | | | | | | | | | |
> | EvoLlama | 25.50 | 25.91 | 22.05 | 27.50 | 14.50 | 24.66 | 29.57 | 23.08 | 20.15 | 27.55 | 27.78 | 21.95 | 24.26 |
> | Evolla | 24.00 | 27.98 | 24.10 | 17.00 | 21.00 | 30.82 | 23.12 | 25.00 | 24.63 | 15.31 | 25.93 | 26.83 | 22.98 |
> | **w task-specific adaptation** | | | | | | | | | | | | | |
> | EvoLlama | 33.00 | 32.12 | 24.10 | 27.00 | 19.00 | 25.34 | 35.48 | 19.23 | 22.39 | 50.00 | 31.48 | 34.15 | 30.00 |
> | Evolla | 26.50 | 25.39 | 23.08 | 23.50 | 25.50 | 19.18 | 26.34 | 19.23 | 20.90 | 15.31 | 29.63 | 26.83 | 23.21 |
> - As shown in the table, task-specific adaptation improves the model's performance.
> - We have incorporated these experimental results and the corresponding analysis into the revised manuscript. Please refer to Appendix D.6 for more details.
>
> **Q1. Qualitative examples (structure).**
> - LLMs reveal different behaviors with or without structural information. They tend to focus more on the secondary structural information instead of the primary sequence information when structural projection images are provided.
> - We have added a qualitative analysis of the structural impact in Appendix E.

---

### Author Response · Authors · 2025-11-26
**General Response**

We thank all the reviewers for the effort engaged in the review phase and the constructive comments.
We have revised our paper (highlighted in blue text color). The modifications are summarized as follows.
1. (For Reviewer t8LX, BW9u). We have added the results of instruction fine-tuning in Appendix D.6.
2. (For Reviewer v6T2). We have added the results of alternative structure encoding in Appendix D.5.
3. (For Reviewer v6T2). We have added the results of compound-protein interaction in Appendix D.7.
4. (For Reviewer t8LX). We have added a qualitative analysis of the structural impact in Appendix E.

---

### Author Response · Authors · 2025-12-01

We sincerely thank the Area Chair and all reviewers for their time and constructive feedback. These reviews have significantly helped us strengthen the quality and depth of our work.

We are encouraged that reviewers recognize the critical value of our work.
- As highlighted by Reviewer t8LX and Reviewer BW9u, LiveProteinBench addresses 'major flaws' and the 'critical issue' of data contamination in existing evaluations.
- The reviewer v6T2 commended the originality and rigor of our 'Live Data' framewor.
- The reviewer BW9u highlighted that our strict post-2025 timeframe 'ensures that none of the test data overlaps with pretraining corpora'.

In response to the reviewers' suggestions, we have made substantial improvements:
- Through one-shot in-context learning experiments, we demonstrated that the performance gap for specialized models is largely an "alignment gap," which can be bridged with minimal tuning. The results are added in Appendix D.6.
- Our ablation study confirmed that 2D projection images outperform text-based 3Di sequences for generalist models, validating our methodological choices. The results are added in Appendix D.5.
- We added Compound-Protein Interaction (CPI) tasks, proving that models struggle significantly with system-level dynamics compared to intrinsic properties. The results are added in Appendix D.7.
- We added detailed case studies explaining instances where structure helped or hindered performance in current MLLMs. The results are added in Appendix E.

We are particularly encouraged by Reviewer v6T2, who acknowledged these revisions and raised his/her score during discussion (from 6 to 8). We hope our extensive new experiments and clarifications can address the concerns raised by other reviewers.

---

### Meta-Review · Area_Chair_DkKc · 2026-01-08

**Summary:**

The paper introduces LiveProteinBench, a contamination-free and multimodal benchmark for assessing large language models’ (LLMs) capabilities in protein science.

The concerns include (1) evaluation focused on zero-shot only (2) justification of using 2D structure projection rather than 3D encoding (3) benchmark focuses on single protein property, omitting interactions and dynamics (4) performance metrics limited to accuracy (5) missing related work (6) lack of methodological innovation and dataset accessibility (7) limited data size (8) paper presentation needs improvement.

**Reviewer Concerns:**

Items 1, 2, 3, 5, 7, 8 are addressed. response to 4, 6 are not convincing.

**Reviewer Scores:**

reviewer v6T2 may raise his score.

---

### Decision · Program_Chairs · 2026-01-26

Reject